# Intercomparison of IBBCEAS, NitroMAC and FTIR for HONO, NO₂ and CH₂O measurements during the reaction of NO₂ with H₂O vapor in the atmospheric simulation chamber of CESAM

Hongming Yi[1]*, Mathieu Cazaunau[2], Aline Gratien[2], Vincent Michoud[2], Edouard Pangui[2], Jean-Francois Doussin[2], Weidong Chen[1]

[1]Laboratoire de Physicochimie de l'Atmosphère, Université du Littoral Côté d'Opale, 59140 Dunkerque, France
[2]Laboratoire Interuniversitaire des Systèmes Atmosphériques, CNRS UMR7583, Universités Paris-Est-Créteil et Université de Paris Diderot, 94010 Créteil, France
*now with Department of Civil and Environmental Engineering, Princeton University, Princeton, NJ 08544, USA

*Correspondence to*: Jean-Francois Doussin (Jean-Francois.Doussin@lisa.u-pec.fr) and Weidong Chen (chen@univ-littoral.fr)

**Abstract.** We report on applications of ultraviolet light emitted diode based incoherent broadband cavity enhanced absorption spectroscopy (UV-LED-IBBCEAS) technique for optical monitoring of HONO, NO₂ and CH₂O in a simulation chamber. Performance intercomparison of the UV-LED-IBBCEAS with a wet chemistry-based NitroMAC sensor and a FTIR spectrometer has been carried out on real time simultaneous measurement of HONO, NO₂ and CH₂O concentrations during the reaction of NO₂ with H₂O vapor in the CESAM atmospheric simulation chamber. 1σ (SNR=1) detection limits of 112 pptv for NO₂, 56 pptv for HONO and 41 ppbv for CH₂O over 120 s were found for the UV-LED-IBBCEAS measurement. On the contrary to many set-ups where cavities are installed outside the simulation chamber, we describe here an original in-situ permanent installation. The intercomparison results demonstrate that IBBCEAS is a very well suitable technique for in situ simultaneous measurements of multiple chemically reactive species with high sensitivity and high precision even if the absorption bands of these species are overlapped. It offers excellent capacity to non-invasive optical monitoring of chemical reaction without any perturbation. For the application to simulation chambers, it has the advantage to provide a spatially integrated measurement across the reactor and hence to avoid point sampling related artefacts.

## 1 Introduction

Atmospheric nitrous acid (HONO) is known as a major source of hydroxyl radicals (OH) (Harris et al., 1982; Finlayson-Pitts et al., 2000) in the atmosphere through its photolysis:

$$HONO + h\nu \ (<400 \text{ nm}) \rightarrow NO + OH \tag{R1}$$

which accounts for 30%-60% of the integrated OH source strength (Alicke et al., 2002; Michoud et al., 2012; Griffith et al., 2016). HONO plays hence a crucial role in the atmospheric oxidation capacity that significantly affects the regional air quality and global climate (Finlayson-Pitts et al., 2000; Stutz et al., 2013). Previous studies have shown that known HONO

sources include heterogeneous reactions, homogeneous gas-phase reactions, direct emission, surface photolysis, and biological processes, respectively (Spataro et al., 2014). HONO formation through the most possible heterogeneous reaction of $NO_2$ with water ($H_2O$) on surfaces is as follows:

$$2\ NO_2 + H_2O \rightarrow HONO + HNO_3 \tag{R2}$$

HONO can be also formed through homogeneous chemistry with the following reaction:

$$NO + OH + M \rightarrow HONO + M \tag{R3}$$

Though it is generally agreed that heterogeneous $NO_2$ chemistry (reaction R2) is probably among the most important sources of HONO (Finlayson-Pitts et al., 2000; Spataro et al., 2014), modelled HONO concentrations are often significantly below observed values (Vogel et al., 2003; Lammel et al., 1996). The sources of HONO and the mechanisms of HONO formation in the troposphere are still under debate (Kleffmann et al., 2007; Sörgel et al., 2011; Li et al., 2014). Although laboratory

studies show that $H_2O$ vapor and surface adsorbed $H_2O$ both play an important role in the conversion process from $NO_2$ to HONO (Finlayson-Pitts et al., 2000; Spataro et al., 2014), the investigations regarding the influence of $H_2O$ on the $NO_2$ and HONO chemistry in the real atmosphere remain a highly discussed topic (Stutz et al., 2004; Michoud et al., 2014) and a well-accepted parameterization is still to come. Scientific questions remain about its sources, sinks, and vertical profile in the

atmosphere (Young et al., 2012; VandenBoer et al., 2013) that will require high precision measurements. In particular, to disentangle the complex mechanisms that are interplaying and affect HONO atmospheric burden, the scientific community needs reliable high frequency assessment of the concentration change of HONO. In both laboratory studies and atmospheric field campaigns, these measurements are challenging due to HONO reactivity and solubility which can cause sampling losses and/or positive artefacts in inlet systems of instruments.

Existing detection methods can be categorized as wet chemistry (WC), mass spectrometry (MS) and optical spectroscopy. In wet chemical methods, HONO is sampled on aqueous/humid surfaces and converted into a species suitable to be analyzed with conventional chemical analytical techniques such as ion chromatography (IC), fluorescence (FL), chemiluminescence (CL), long-path absorption photometer (LOPAP) or high-performance liquid chromatography (HPLC) (Chen et al. 2013). These wet-chemical-based instruments often suffer from unquantified chemical interferences and

sampling artifacts (Stutz et al., 2010). Moreover, calibrations of the instruments based on WC and MS are difficult, because no permanently stable calibration mixtures exist for HONO.

Intercomparison of ambient HONO measurement instruments have been carried out between differential optical absorption  spectroscopy (DOAS) and long path absorption photometer (LOPAP) (Kleffmann et al., 2006); between DOAS, mist-chamber/ion chromatograph (MC/IC), stripping coil-visible absorption photometry (SC-AP), ion drift-chemical

ionization mass spectrometry (ID-CIMS), and quantum cascade-tunable infrared laser differential absorption spectroscopy (QCL-TILDAS) (Pinto et al., 2014); between LOPAP and NitroMAC (French acronym for "continuous atmospheric measurements of nitrogenous compounds") (Afif et al., 2016); between LOPAP and incoherent broadband cavity enhanced absorption spectroscopy (IBBCEAS) (Wu et al., 2014); between LOPAP, Fourier transform infrared spectrometer (FTIR) and differential photolysis (Reed et al., 2016). Quite frequently, intercomparison between in point and long-path

measurements exhibited significant discrepancies with uncertainties within about 20% (Pinto et al., 2014; Kleffmann et al., 2006) for HONO concentrations from ten-pptv to ten-ppbv range.

Calibration-free, high-sensitivity, direct HONO measurement with UV-IBBCEAS is capable of providing accurate and fast quantitative analysis of HONO concentration variation within its lifetime, which is crucial to improve the understanding of the atmospheric behaviour of HONO. Although the main interest for current work is to measure HONO, $NO_2$ and $CH_2O$

are two other important atmospheric species (Washenfelder et al., 2016; Liu et al., 2020), these two molecules have strong absorption in the same region. Simultaneous measurements and quantification of HONO, $NO_2$ and $CH_2O$ can be performed by the IBBCEAS techniques (Wu et al., 2014; Washenfelder et al., 2016; Duan et al., 2018; Jordan et al., 2020).

In the present work, we report on the development of an ultraviolet light emitted diode (UV-LED) based UV-IBBCEAS instrument for simultaneous measurement of concentrations ranges of HONO (0-30 ppbv), $NO_2$ (0-120 ppbv) and $CH_2O$ (0-

150 ppbv) during the processes of HONO generation through $NO_2$ reaction with $H_2O$ in a simulation chamber. HONO, $NO_2$, $CH_2O$ and $H_2O$ vapor concentrations were real time tracking at well controlled conditions. Intercomparison measurements of HONO concentration by UV-IBBCEAS vs. NitroMAC and UV-IBBCEAS vs. FTIR; $NO_2$ concentration by UV-IBBCEAS vs. chemiluminescence and UV-IBBCEAS vs. FTIR were addressed during a 3-day campaign in the atmospheric simulation chamber CESAM. In addition, Intercomparison measurements of $CH_2O$ concentration by UV-IBBCEAS vs. FTIR was also

simultaneously committed at the last 8 hours of the third day. Agreement of uncertainties <10% were acquired for $NO_2$, HONO and $CH_2O$.

## 2 Experimental details

### 2.1 Intercompared instruments

#### 2.1.1 LED based UV- IBBCEAS set-up

The LED based UV-IBBCEAS setup installed in the simulation chamber, which was used for measurements of $NO_2$, HONO and $CH_2O$ concentrations in the present work, is shown in Fig. 1. A UV-LED (Nichia, NCSU033AT), emitting ~300 mW optical power and with divergence angle of ±60º in the UV spectral region around 365 nm was used as probing light source. The LED source was mounted on a temperature-controlled heat sink made of copper block to stabilize the output optical intensity and spectral profile of the LED emission. The temperature of the copper plate was stabilized at 20 ºC within ±0.01

ºC by means of a single-stage thermo-electric cooler (TEC, PE-063-08-15, Supercool) associated with a temperature sensor (PT100, RTD). A laser diode controller (LDC501, Stanford Research System) was used to supply electric power for both the TEC and the UV-LED. A high-finesse optical cavity was formed with two high-reflectivity mirrors (Layertec GmbH) that were installed in the simulation chamber walls facing each other (Fig. 2), separated by the diameter of the cylindrical CESAM chamber, $L$=2.13±0.05 m. The cavity mirrors had 25 mm in diameter, 2 m radius of curvature and 6.35 mm

thickness. The experimentally measured reflectivity of the mirrors is shown in Fig. 3(a) between 350 and 380 nm with a peak

value of R~99.95% at 360 nm. The enhancement factor of the cavity is wavelength-dependent $F=1/(1-R(\lambda))$, ranging from $F=2000$ at $\lambda= 365$ nm to $F=1250$ at $\lambda=378$ nm, corresponding to equivalent absorption path lengths through the intra-cavity sample between 4.2 and 2.6 km. Light from the LED was focused with achromatic lens L2 (BK7, $f=75$ mm) into the optical cavity. In order to avoid CCD spectrometer saturation at the edges of high reflectivity range of the cavity mirrors, a band-pass filter (Semrock 340-390 nm) was placed between achromatic lens L2 and the cavity to block the light at undesirable wavelengths. The diameter (~10 mm) of the light beam injected into the cavity was controlled with an iris. The light transmitted through the cavity was collected through achromatic lens L3 (BK7, $f=75$ mm) to a multimode optical fiber (1000 μm in diameter) and coupled to a CCD spectrometer (QE65000, Ocean optics). A thermoelectric cooler (TEC) was used to cool CCD-camera temperature to 40 °C below ambient temperature to avoid wavelength drifts as well as to remove dark noise and readout noise. The spectrometer allowed covering the whole 190–480 nm spectral range with a spectral resolution of 0.59 nm around 365 nm (this spectral resolution is sufficient for selective recognition of the structured broadband absorption of $NO_2$, $CH_2O$ and HONO). The measured spectra from the spectrometer were recorded by a laptop computer through a USB interface.

### 2.1.2 Wet chemical technique

NitroMAC is an analytical instrument developed for field measurement of atmospheric HONO. Based on the original work of (Huang et al., 2002), the concept of NitroMAC relies on a wet chemical derivatization and detection of absorption in the visible at 540 nm using high performance liquid chromatography (HPLC). The instrument has been described in detail in another reference (Afif et al., 2016), but in the present study the instrument was equipped with a dedicated external sampling unit similar to the one of LOPAP instrument (Villena et al., 2011) to minimize potential artefact in the sampling line. Sample gas from the simulation chamber is pumped into NitroMAC with a flow rate of 2 L/min. HONO is sampled in a temperature-controlled stripping coil by a fast-chemical reaction in the stripping reagent and a few centimeters (ca. 5 cm) from the chamber port. It is right away converted by dissolution in a buffer phosphate solution and followed by derivatization of nitrite to a highly light-absorbing azo-dye with sulfanilamide (SA) and N-(1-naphthyl)-ethylenediamine (NED) and then transferred to the analytical unit. The operation mode for this instrument consists of two coils connected in series. The arrangement of the two identical coils in series allows the determination of sampling efficiency or the evaluation of possible interferences in HONO measurements. The response obtained by integration of the chromatographic peak for the second stripping coil 1 is then subtracted from that of the first one to eliminate interferences. HONO concentrations are then calculated from this net signal using calibration factors determined through direct calibrations of the analytical system (HPLC-UV-Visible) performed using $NaNO_2$ standard solutions. The performance of this instrument in terms of its detection limit was found to be around 3 pptv with an optimal integration time of 10 min. The relative standard deviation is 2%, and the relative expanded measurement uncertainty is 12% with a signal to noise ratio of 2 (2σ) (Michoud et al., 2014).

### 2.1.3 FT-IR spectrometer

A Fourier Transform Infra-Red (FTIR) spectrometer equipped with a White type multipass cell was used. Its main purpose was to calibrate the mirror reflectivity in the IBBCEAS setup based on concentration measurements of $NO_2$. This spectrometer (model: Bruker® Tensor 37TM) is equipped with a liquid nitrogen cooled MCT detector and a globar source. The multipass cell consists of three high reflectivity gold coated mirrors with a base length of 1.9 m. The configuration of this White cell provided 96 reflections between three mirrors, and offered a total optical path length of 182±1 m. The multiple path system was crossing the chamber in the same plane as the IBBCEAS pathway with an angle of 60° between the two main optical axis (see Fig. 1). The FTIR system records spectra in the infrared range between 500 and 4000 $cm^{-1}$ with an optimal resolution of 0.5 $cm^{-1}$. A typical experiment leads to the acquisition of hundreds of FTIR spectra. To perform the analysis of huge datasets, a home-made software algorithm was written to retrieve $NO_2$ concentrations. Typical detection limits in absorption spectra recorded by co-adding 100 scans (i.e. with an integration time of 5 min) for various gaseous compounds are listed as follows: $NO_2$ (5 ppbv typically but here 20 ppb because of the use of smaller absorptions regions), $O_3$ (5 ppbv), HONO (10 ppb), $CH_2O$ (3 ppbv) or $HNO_3$ (10 ppbv).

### 2.1.4 NOx analyzer (Chemiluminescence)

In the current experiment, $NO_2$ is measured using a chemiluminescence (CL) NOx (=NO+$NO_2$) analyzer (Horiba, model APNA360) equipped with a molybdenum converter (Sigsby et al., 1973). $NO_2$ was indirectly measured by first converting it into NO and measuring the sum of NO + $NO_2$. $NO_2$ is transformed to NO via a heated converter using molybdenum, and the $NO_2$ concentration is obtained as the difference between the NO-only measurement and the NO + $NO_2$ measurement. Chemiluminescence instruments are typically calibrated with a NO mixture, usually in $N_2$, which is injected directly or converted to $NO_2$ via gas-phase titration (Tidona et al., 1988). It is well known that these instruments are subject to strong positive interferences from NOy (Dunlea et al., 2007) as a large class of nitrogenous compound may be converted on heated Mo converters to produce NO and lead to a chemiluminescence signal on the $NO_2$ channel. For HONO in particular this interference is considered being quantitative (Villena et al., 2012). In our chemical system, HONO is expected to be the main NOy species interfering with the $NO_2$ measurement, its concentration when available from NitroMAC where subtracted from the APNA 360 "$NO_2$ channel" to provide "corrected" $NO_2$ concentration and assuming 100% conversion efficiency for HONO.

### 2.1.5 Temperature and humidity sensor

Temperature and relative humidity (RH) inside the simulation chamber were recorded with a temperature and humidity sensor (Vaisala HMP 234) (T&RH sensor). Absolute water vapor concentrations were calculated using the measured RH, the corresponding temperature and the pressure. The measurement error is 1% for RH and 0.1 °C for temperature at atmospheric pressure and room temperature.

## 2.2 Intercomparison experiments and set-up

The CESAM simulation chamber is a 4.2 m³ stainless steel chamber. It has been described in detail elsewhere (Wang et al., 2011) and only key information will be recalled here. The CESAM simulation has roughly a cylindrical shape with a 1.7 m diameter. When adding the length of the flanges that support the various inlets and instruments (see Fig. 2), it provides a 2 m long diameter that is exploited here to provide the unitary pass length of both the FTIR and IBBCEAS analytical pathway.

The intercomparison set-up is shown in Fig. 1. The measurement instruments, such as the custom-made UV-LED-IBBCEAS and NitroMAC, a chemiluminescence (CL) NOx analyzer (HORIBA APNA 370) and a FTIR spectrometer (Bruker Tensor 37), are installed around the atmospheric simulation chamber.

The experiments were performed at room temperature and atmospheric pressure (~23 ℃ and 760 Torr). Firstly, the chamber was cleaned by pumping down to secondary vacuum (ca. $10^{-4}$ mbar). The chamber was then filled with clean dry air by mixing 800 mbar of nitrogen produced from the evaporation of a pressurized liquid nitrogen tank (Messer, purity > 99.995 %, $H_2O$ < 5 ppmv), and 200 mbar of oxygen (Air Liquide, ALPHAGAZ™ class 1, purity 99.9 %). The mixture was left ca. 45 minutes for the acquisition of recording instrument background. 500 µl of gaseous $NO_2/N_2O_4$ mixture was then introduced with a gas-tight syringe (from an $NO_2$ cylinder: Air Liquide™, Alphagaz™ 99.9% purity) leading to about 120 ppbv of $NO_2$ in the CESAM chamber. A small pressurize stainless steel vessel filled with ultrapure water (18.2 Mohm, ELGA Maxima) was used to produce the required water vapor. When the $NO_2$ concentration inside the chamber was stabilized at 120±5 ppbv, the prepared $H_2O$ vapor was introduced into the simulation chamber. The relative humidity (RH) inside the chamber was allowed to increase to ~66% at 23 ℃ (corresponding to an absolute $H_2O$ vapor mixing ratio of ~1.85%). There were 4 experiments during the whole measurement, the 2[th] to 4[th] experiments were performed under the same experimental conditions as the first one. The four experiments were followed by the same procedure.

Under these conditions, as described by Wang et al. (2011), the desired amount of gas-phase HONO is systematically observed. As stated in the literatures (Finlayson-Pitts et al., 2000; Lammel et al., 1995; Spataro et al., 2014), HONO is generated through heterogeneous formation on the chamber's inner surfaces via a complex reaction of $NO_2$ with $H_2O$ adsorbed on the chamber walls. All instruments (UV-IBBCEAS, FTIR, NitroMAC, NOx analyzer, temperature and humidity sensor, pressure gauge) simultaneously recorded the relevant data (including $NO_2$, HONO, NO and $H_2O$ concentrations, temperature and pressure) for data analysis and instrument intercomparison. Absolute $NO_2$ concentrations obtained by the FTIR were used to determine cavity mirror reflectivity.

Four $NO_2$ injections in the presence of humid air were organized during the four days of experiments. During the last experiment, an injection of formaldehyde (HCHO) was performed to allow the investigation of the sensitivity of the UV-IBBCEAS data analysis to the interferences in the UV range. Formaldehyde was prepared by sublimating commercial paraformaldehyde $(CH_2O)_n$ (Fluka, "extra pure" grade) under vacuum in a glass line and collected at a known pressure in a bulb of known volume. This quantity was then flushed into the chamber with a gentle flow of pure nitrogen. A controlled

dilution flow was allowed to the chamber to induce a forced decrease of the sampled concentrations and hence testing the quantification performance of the various analytical devices across a few orders of magnitudes.

## 2.3 Data analysis

### 2.3.1 UV-LED-IBBCEAS

In an IBBCEAS experiment, the transmitted spectra $I_0(\lambda)$ from the cavity without absorbing species are firstly measured by

filling the cavity with pure $N_2$ or zero air, and then the spectra $I(\lambda)$ in the presence of target sample are recorded. The absorption by molecular species, Rayleigh scattering by molecular species $\alpha_{Ray}(\lambda)$, Mie scattering by particles $\alpha_{Mie}(\lambda)$ and absorption by particles $\alpha_{abs-particle}(\lambda)$ contribute to optical light extinction in the cavity, the total optical extinction coefficient $\alpha(\lambda)$ is given by (Gherman et al., 2008; Fuchs et al., 2010; Wu et al., 2012; Wu et al., 2014; Duan et al., 2018; Jordan et al., 2020):

$$\alpha(\lambda) = \left( \frac{1-R(\lambda)}{d} + \alpha_{Ray}(\lambda) + \alpha_{Mie}(\lambda) + \alpha_{abs-particle}(\lambda) \right) \times \left( \frac{I_0(\lambda)}{I(\lambda)} - 1 \right) \quad\quad (1)$$

where $d$ is the distance between two cavity mirrors. Here $\alpha_{Ray}(\lambda)$, $\alpha_{Mie}(\lambda)$ and $\alpha_{abs-particle}(\lambda)$ are needed to consider for real atmospheric condition or open-path observation. However, $\alpha_{Mie}(\lambda) \approx 0$ and $\alpha_{abs-particle}(\lambda) \approx 0$ can be neglected for particle-free environment.

    In the present work of gas-phase chemical reaction in the simulation chamber filled by zero air, the chamber is free of

particles, thus $\alpha_{Mie}(\lambda) \approx 0$ and $\alpha_{abs-particle}(\lambda) \approx 0$. Otherwise, low-concentration $NO_2$ in air (<200 ppbv) was used for mirror reflectivity $R(\lambda)$ determination, the Rayleigh scattering coefficient of zero air ($\alpha_{Ray}(\lambda)$ in Eq.1) between 350 nm and 380 nm ($\alpha_{Ray-Zero\,air} \sim 10^{-8}\,cm^{-1}$) can be neglected, $\alpha_{Ray}(\lambda) \approx 0$, thus $R(\lambda)$ can be determined by using a known-concentration $NO_2$ sample as below:

$$R(\lambda) = 1 - d \left( \alpha_{NO_2} \times \frac{I_{NO_2}(\lambda)}{I_{Zero\,air}(\lambda) - I_{NO_2}(\lambda)} \right) \quad\quad (2)$$

where $I_{NO2}$ and $I_{zero\,air}$ are the transmitted LED light intensities through the cavity containing $NO_2$ and zero air, respectively, $\alpha_{NO2}$ is the absorption coefficient of $NO_2$ and is. In order for determination of $R(\lambda)$, about 100-200 ppbv $NO_2$ (absolute $NO_2$ concentrations were determined by the FTIR spectrometer) was injected in the simulation chamber. Using the known $NO_2$ concentrations measured in-situ by the FTIR spectrometer and Rayleigh scattering cross section of zero air (see Fig. 3(b)), the mirror reflectivity can be deduced from Eq. (2), as shown in Fig. 3(a). With the measured mirror reflectivity $R(\lambda)$, the

mirror-to-mirror distance of optical cavity $d$, and absorption cross sections $\sigma(\lambda)$ of the target gas from a common database, target gas concentrations can be simultaneously retrieved using a least-squares fit to the experimentally measured absorption coefficient $\alpha(\lambda)$:

$$\alpha(\lambda) = \frac{1-R(\lambda)}{d} \times \left( \frac{I_0(\lambda)}{I(\lambda)} - 1 \right) = n_{NO_2} \cdot \sigma_{NO_2}(\lambda) + n_{HONO} \cdot \sigma_{HONO}(\lambda) + n_{CH_2O} \cdot \sigma_{CH_2O}(\lambda) + a\lambda^2 + b\lambda + c$$

(3)

where $\sigma_{NO2}(\lambda)$, $\sigma_{HONO}(\lambda)$ and $\sigma_{CH2O}(\lambda)$ are the reference absorption cross sections (in [$cm^2$/molecule]) of $NO_2$, HONO and

$CH_2O$ species (see Fig. 3(b)), respectively. As shown in Fig. 3(a), the chosen UV-LED emission covers an absorption band

of 350-380 nm including $NO_2$, HONO, and $CH_2O$ contributions. The reference cross sections of $NO_2$ (Voigt et al., 2002),

HONO (Stutz et al., 2000) and $CH_2O$ (Meller et al., 2000) were convoluted with the instrument function of approximately

0.59 nm (the spectrometer resolution). $n_{NO2}$, $n_{HONO}$ and $n_{CH2O}$ are the concentrations (number densities) of $NO_2$, HONO and

$CH_2O$, respectively. The second-order polynomial term in Eq. (3) represents the variation in spectral baseline which could

arise from gas scattering, LED intensity fluctuations, and other unspecified loss processes. The unknown parameters

(number densities $n_{NO2}$, $n_{HONO}$, $n_{CH2O}$, $a$, $b$ and $c$) can be extracted using a linear algebraic method known as the singular

value decomposition (SVD) method (Varma et al., 2009; Yi et al., 2016). A Labview based concentration retrieval program

was used to simultaneously process the data to provide real-time $NO_2$, HONO and $CH_2O$ concentrations.

Acquisition time for each spectrum was 2 minutes, the statistical error of each individual spectrum is close to ~1%. This

~1% statistical error is as good as the results reported in the references for other IBBCEAS setups (Kleffmann et al., 2007;

Fuchs et al., 2010; Varma et al. 2009; Gherman et al., 2008; Rodenas et al., 2013; Min et al., 2016). The measurement

uncertainty in the retrieval of trace gas mixing ratios are dominated by the uncertainties in the used absorption cross-sections

of HONO, $NO_2$, and $CH_2O$ ($\pm5\%$, $\pm3\%$, and $\pm3\%$, respectively) (Voigt et al., 2002; Stutz et al., 2000; Meller et al., 2000), in

the determination of $(1-R)$ (~7%), in the measurement of $I_0/I$ (0.5%), and in the cavity length determination (<1%). The

total relative uncertainty in the retrieved concentrations, including the statistical uncertainty from the fit (<0.5%) and the

measurement uncertainty, is approximately estimated to be ~9 % for HONO, ~8% for both $NO_2$ and $CH_2O$, respectively.

     Typical UV-IBBCEAS spectra (from 351 to 378 nm) of 18.0 ppbv HONO, 93.3 ppbv $NO_2$ and 143 ppbv $CH_2O$ as well

as the total fit for their concentration retrievals are given in Fig. 4(a) (black line). In order to well indicate individual

absorption for single molecule, the decomposed spectral (Kennedy et al., 2011) associated with the corresponding fits for

HONO, $NO_2$ and $CH_2O$ were shown in Fig. 4(a) (blue line), Fig. 4(b) (black line), and Fig. 4(b) (green line), respectively.

Based on the fit residual, the corresponding $1\sigma$ minimum detectable concentration (MDC) with mixing ratio for 120 s

integration time are 112 pptv for $NO_2$, 56 pptv for HONO using 362-372 nm region data. MDC for $CH_2O$ with 120 s is 41

ppbv by using of 351-360 nm spectral data.

Allan variance analysis was carried out to assess the stability (corresponding to the optimal integration time) of the UV-

IBBCEAS setup. Zero air was used to purge the simulation chamber. Time-series spectra of zero air were recorded with a

rate of 1 spectrum per second, about 2000 spectra were acquired for the Allan variance study (Wu et al., 2012; Yi et al.,

2015). Typical Allan variance curves are plotted in Fig. 5, illustrating a highly desired white noise dominated system

stability. As a compromise between detection limit (requiring long integration time) and measurement time response

(requiring short measurement time), an integration time of 120 s was selected for use in the present work, which correspond to the measurement precision of 100 pptv for $NO_2$, 30 pptv for HONO and 40 ppbv for $CH_2O$.

### 2.3.2 FTIR spectra

Infrared spectra were obtained at a resolution of 0.5 $cm^{-1}$ and derived from the co-addition of approximately 200 scans collected over 5 min. Each scan was obtained from the Fourier transform of an interferogram apodized with the Happ-
Genzel function. Concentrations of the target species were determined by subtracting pure reference spectra (brought to the experimental resolution of 0.5 $cm^{-1}$) from spectra of reaction mixtures using home-made software based on matrix algebra. To guarantee the performance of the automatic routine, selected spectra for each experiment were subtracted manually and results were compared. Spectroscopic information used for the FTIR data analysis are given in table 1. HONO absorption was analyzed from its $v_3$ absorption bands around 1263 $cm^{-1}$ using the synthetic reference spectrum proposed by Barney et
al. (2000) and modified by Barney et al. later (Barney et al., 2001). FTIR spectra of mixture where analyzed using the ANIR deconvolution software (Rodenas et al., 2020), which uses a linear square fitting method to quantitatively analyze experimental spectra through a combination of reference spectra.

### 3 Results and discussion

During the intercomparison experiments in the CESAM atmosphere simulation chamber, time series measurements of $NO_2$,
HONO and $CH_2O$ were simultaneously performed using the UV-LED-IBBCEAS, FTIR spectrometer, NOx analyzer and NitroMAC.

Before HONO, $NO_2$ and $CH_2O$ intercomparison measurements, the FTIR spectrometer was used to measure absolute $NO_2$ concentrations between 60 and 120 ppbv in order to determine of the wavelength-dependent reflective curve of cavity mirror. The measured mirror reflectivity ($R$) is shown in Fig. 3(a) (red line). During the HONO generation process, $NO_2$ and
$H_2O$ vapor were introduced into the simulation chamber four times, which correspond to four peaks of $NO_2$ (as shown in Fig. 7(a)). The maximum $H_2O$ vapor concentrations measured by T&RH sensor are 1.85%, 1.54% and 1.63% for 2nd to 3rd peaks, respectively. At the 4th peak in Fig. 7(a), ~160 ppbv $CH_2O$ was also introduced into the chamber to evaluate the UV-LED-IBBCEAS performance for simultaneous detection of $NO_2$, HONO and $CH_2O$. This process explains the peak shape formed of a straight injection step followed by an exponential decay during four-day experiments (1st to 4th peaks in Fig.7 (a) and
Fig. 8 (a)).

The UV-LED-IBBCEAS spectrometer recorded spectra of the transmitted light intensity with an integration time of 1 s, and 120 data acquired in this manner were then averaged to produce one spectrum for $I_0(\lambda)$ or $I(\lambda)$ (i.e. a net acquisition time of 2 min per spectrum). The integration time of the NitroMAC was 10 min for one measurement of HONO concentration. The integration time of the FTIR for $NO_2$, HONO, $CH_2O$ and $H_2O$ vapor was 1 min.

## 3.1 Side-by-side comparison of NO₂ and HONO measurements

Intercomparison of HONO measurements were executed between the UV-LED-IBBCEAS, the NitroMAC and the FTIR, while the measured $NO_2$ concentrations were compared between the UV-LED-IBBCEAS, the NOx analyzer and the FTIR.

For in-situ $NO_2$ monitoring, the correlation between NOx analyzer and IBBCEAS measurements is not linear. The NOx analyzer over estimated $NO_2$ concentrations during all measurements, as shown in Fig. 6(a), which was caused by the well-known positive interferences (overestimation) (Villena et al., 2012) in the $NO_x$ analyser due to non-selective conversion of all nitrogen containing species inside the chamber into NO (Tidona et al., 1988; Villena et al., 2012) for the indirect measurement of $NO_2$ concentrations. In the present experiment, the main interferences source was HONO that was transferred into NO in the NOx analyser, which resulted in an overestimation of the $NO_2$ concentration. The amount of the overestimated $NO_2$ concentration equal to HONO concentration simultaneously measured by NitroMAC in the current study. The real $NO_2$ concentration can be obtained by deduction of HONO concentration simultaneously measured by NitroMAC from $NO_2$ concentration measured by NOx analyser. After correction of the HONO contribution to the measured $NO_2$ concentrations, time series intercomparison measurements of $NO_2$ between the NOx analyzer (with HONO correction) and the UV-LED-IBBCEAS are shown in Fig. 6(b), which shows a good agreement between the two instruments. Measurements of $NO_2$ have been then compared between the UV-LED-IBBCEAS, the NOx analyzer (with HONO correction) and the FTIR, as shown in Fig. 7(a). $NO_2$ concentrations ranging from 100 pptv to 140 ppbv were investigated during the entire experimental intercomparisons, the corresponding correlation analyses are plotted in Figs. 7(b,c). A linear correlation coefficient of $r^2$=0.987 was acquired between data from the interference-corrected NOx analyzer and the IBBCEAS instrument (Fig. 7(b)), both measurements agree well (slope=1.051) with an offset of 0.130 ppbv. The plot of $NO_2$ measurements by FTIR vs IBBCEAS shown in Fig. 7(c) presents a linear correlation with a $r^2$=0.885, the fitted slope and offset are 0.933 and 0.265 ppbv, respectively. This discrepancy of about 7% between FTIR and IBBCEAS mainly comes from the larger relative measurement uncertainty of the FTIR due to its worse MDC of 10 ppbv at sampling time of 5 min compared to that of 112 pptv with 2-min integration time for IBBCEAS.

Time series intercomparison measurements of HONO by UV-LED-IBBCEAS, NitroMAC and FTIR are shown in Fig. 8(a). To provide a more quantitative intercomparison, a linear regression analysis was performed, weighted with errors of three instruments (IBBCEAS vs. NitroMAC and IBBCEAS vs. FTIR). The comparison of all data and the results of the regression analysis are shown in Figs. 8(b,c). From these results, the HONO concentrations measured by the three instruments display the same variation trend when the HONO concentration varied from 0 to 40 ppbv (2nd-4th peaks in Fig. 8(a)). For the region of 2nd and 3rd peaks in Fig. 8(a), HONO concentrations from NitroMAC are 33% and 35% higher than that from IBBCEAS, respectively. During the 4th HONO generation process (peak 4) in Fig. 8(a), the correlation between the NitroMAC and the IBBCEAS becomes better, NitroMAC measurement is only 8.4% higher than that from IBBCEAS instrument. In this HONO generation process, about 150 ppbv $CH_2O$ was injected into the chamber (Fig. 9(a)). Nevertheless, it is not possible to relate this better correlation result to the presence of formaldehyde. It is hypothesized that the speed of

the mixing fan was increased during the last part of the experiment, and by improving the mixing, the point measurement by NitroMAC nearby the walls are getting more comparable with the spatially integrated value from the IBBCEAS. The correlation between these two instruments during the entire experiment is $r^2=0.954$ (Fig. 8(b)), the gradient of this weighted regression is 1.273 with a y-axis intercept of 0.067 ppbv between the NitroMAC and the IBBCEAS (Fig. 8(b)), showing an overall level of agreement within 27% throughout the entire experiment. Considering the relative measurement uncertainty of 12% for NitroMAC and 9% for IBBCEAS (a total uncertainty of 21% for two-instrument system), this difference is close to the measurement errors. A small systematic discrepancy is nevertheless remaining after the uncertainties analysis. It is hypothesized that this disagreement may arise from the sampling volumes of the two techniques and of the HONO generation mechanism. First, IBBCEAS (similarly to FTIR) is providing a spatial average of the concentration across the chamber while NitroMAC is a single point sampler located at the bottom of a side port (see Fig. 1) ca. 20 cm away from the main well-mixed chamber volume. Further, HONO generation is a multiphase process that involves wall reaction. The local wall-to-volume ratio around the NitroMAC inlet is certainly larger than the average wall-to-volume ratio of the CESAM. This may explain why in most of the cases NitroMAC values were larger than those measured by IBBCEAS. On the other hand, in the IBBCEAS, the final HONO concentrations depend on the selected HONO cross sections (Gratien et al., 2009), the HONO time-concentration profiles in Fig. 8(a) were retrieved using the absorption cross section published by Stutz et al., 2000. If the absorption cross section from another publication (Brust et al., 2000) was used to retrieve HONO concentration, all HONO concentrations in IBBCEAS will increase 23%, which equal to multiply a factor of 1.23 to the currently presented HONO concentrations in Fig. 8(a). In this case, good agreement (with a linear-fit slope approaching 1) is observed between the HONO concentrations measured by LED-IBBCEAS and NitroMAC, respectively.

The correlation and the regression analysis for the comparison between the FTIR and the IBBCEAS ($2^{rd}$-$4^{th}$ peaks) is given in Fig. 8(c), displaying a slope of 0.952 with a y-axis intercept of 0.250 ppbv and a $r^2=0.89$. HONO-concentration variation profile ($2^{rd}$-$4^{th}$ peaks in Fig. 8(a)) coincides well with each other between IBBCEAS and FTIR with a correlation slope close to 1. The discrepancy (<5%) is mainly due to the larger measurement uncertainty of HONO by FTIR. FTIR used the integrated HONO absorption band intensity to retrieve HONO concentration, interference from other species is hard to avoid, such as $NO_2$, $HNO_3$ and $H_2O$ absorption in the 1200-1300 $cm^{-1}$ region.

### 3.2 Interferences and opportunity for formaldehyde measurements using IBBCEAS

Formaldehyde is ubiquitous in the atmosphere and is among the most probable interfering species for the deployment of the UV-LED-IBBCEAS as it exhibits strong absorptions between 260 and 360 nm. It is thus important: (a) to investigate any potential artifact during its co-detection with HONO and (b) to evaluate through intercomparison the ability of the newly developed IBBCEAS to reliably quantify it.

During the $NO_2$ and HONO intercomparison campaign around the $4^{th}$ peak region, about 150 ppbv $CH_2O$ was added into the chamber in order to evaluate potential interference to the IBBCEAS data analysis. The $CH_2O$ concentrations ranging from 0

to 150 ppbv were investigated using UV-LED-IBBCEAS and FTIR, the time series measurements are plotted in Fig. 9(a). A good linear correlation between the measurements by two instruments is obtained with a regression slope of 1.053 and an intercept of 3.653 ppbv ($r^2$=0.971), as shown in Fig. 9(b). This measurement intercomparison confirmed the good performance of the measurement of $CH_2O$ using IBBCEAS. The relatively large intercept of 3.653 ppbv is due to the relatively high MDC of 41 ppbv because the used UV-LED emission intensity was very weak at its side wing near 350 nm at

which $CH_2O$ was probed (Fig. 2) which degraded significantly the SNR (signal to noise ratio) in the IBBCEAS spectrum of $CH_2O$. Moreover, the corresponding $CH_2O$ absorption cross section near 350 nm is not the maximal value in this region for its sensitive measurement. The MDC can be further improved by using a suitable light source with main emission centered between 315-350 nm allowing to probe the strongest $CH_2O$ absorption lines which may lead to a MDC of 0.38 ppbv (Washenfelder et al., 2016; Liu et al., 2020). The present work, with excellent measurements correlation on $NO_2$, HONO and

$CH_2O$ between IBBCEAS and other well-established instruments, shows that the IBBCEAS technique offers the ability of self-calibration based on unique wavelength-dependent specific absorption intensity of the target molecules for simultaneously measuring concentrations of these three species with high precision without significant interference influence even if their absorption cross sections are overlapped. The present work in an atmospheric simulation chamber, with excellent measurements correlation on $NO_2$, HONO and $CH_2O$ between IBBCEAS and other well-established instruments,

shows that the IBBCEAS technique offers the ability of self-calibration for simultaneously measuring concentrations of these three species with high precision without significant interference influence even if their absorption cross sections are overlapped. For its application to an uncontrolled environment, the interference resulting from the presence of aerosols, in particular, would degrade the performance of the IBBCEAS measurement which is an issue to be carefully addressed. Under harsh environmental conditions, additional approaches, such as purging high-reflectivity mirror, using particle filter to

reduce aerosol absorption and scattering, could be associated to extend the IBBCEAS technique to field campaign (Wu et al., 2014; Duan et al., 2018; Jordan et al., 2020).

## 4 Conclusion

Intercomparison measurements of HONO, $NO_2$ and HCHO between IBBCEAS, NitroMAC and FTIR have been performed

during the reaction of $NO_2$ with $H_2O$ vapor in the CESAM atmosphere simulation chamber. The performance of IBBCEAS was evaluated through side-by-side comparison with NitroMAC and FTIR for HONO, with FTIR and NOx analyzer for $NO_2$, and with FTIR for $CH_2O$. The intercomparison of the measured data shows a good agreement on the temporal trends and variability in HONO, $NO_2$ and $CH_2O$. Good correlation of better than 93% for $NO_2$ measurements between IBBCEAS, NOx analyzer and FTIR was obtained under a well-controlled condition in the CESAM simulation chamber. Due to positive

interference, $NO_2$ concentration measurement using NOx analyzer was corrected by deduction of HONO contribution. A more than 95% correlation for $CH_2O$ measurements was found between IBBCEAS and FTIR. The measured time-series HONO profiles displayed a relatively large divergence (up to 30%) in absolute concentrations from the intercomparison between IBBCEAS and NitroMAC. NitroMAC indicated somewhat higher HONO concentration than those from the IBBCEAS and the FTIR. This discrepancy of ~27% can only be partly attributed to the uncertainty of the cross sections used for HONO concentration retrieval. A significant fraction of the discrepancy can most probably be attributed to the fact that NitroMAC was sampling in a point that is relatively protected from mixing fan effects and close to the wall i.e. where HONO is being produced. This drawback of our experimental strategy did not harm too seriously our assessment of the IBBCEAS set-up and retrieval thanks to the use of in-situ FTIR, which had the advantage to illustrate well how important it is to perform measurements that spatially depends on the probed volume. It illustrates how in-situ spatially averaged measurements are the strategy of choice for the monitoring of reactive species in simulation chambers.

The experimental results and relevant analysis show that UV-LED-IBBCEAS has advantages of studying chemical dynamics by means of in situ and fast concentration tracking with high precision. It also has the capacity of simultaneously and directly measuring $NO_2$, HONO and $CH_2O$ in chamber experiment without any sample extraction and hence without any influence on the chemical reaction going on, which offers a unique advantage of non-invasive monitoring of chemical reaction in chamber studies. Its absorption line intensity based on self-calibration capacity exhibits another advantage compared to the need of complicated calibration process using chemical solutions for wet chemistry based analytical instrument.


**Data availability**

The data used in this study are available from the corresponding author upon request (Jean-Francois Doussin (Jean-Francois.Doussin@lisa.u-pec.fr) and Weidong Chen (chen@univ-littoral.fr)).

**Author contributions**

The manuscript was written through contributions of all authors. All authors have given approval to the final version of the manuscript.

**Competing interests**

The authors declare that they have no conflict of interest.

**Acknowledgments**

The authors acknowledge financial supports from the French Agence Nationale de la Recherche (ANR) under the CaPPA (ANR-10-LABX-005) contract, as well as the support in the framework of the CPER CLIMIBIO program funded by Nord-

Pas de Calais Region and the Ministère de l'Enseignement Supérieur et de la Recherche. CNRS-INSU is gratefully acknowledged for supporting the CESAM chamber as a national facility (*http://cesam.cnrs.fr*). European Union's Horizon 2020 research and innovation program through the EUROCHAMP-2020 Infrastructure Activity under grant agreement n°730997 is also gratefully acknowledged for distributing freely the ANIR deconvolution software as well as Mila Rodenas and the Centro de Estudios Ambientales del Mediterráneo (CEAM - Valencia) for sharing the software and providing the

guidance in using it.

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

**Table 1** Absorption bands adopted by FTIR for $NO_2$, HONO, $CH_2O$ and $H_2O$ measurement. IBI: Integrated Band Intensities

| Species | IBI, base log10 | Integration borders (cm⁻¹) | Spectrum origin / References | Spectral windows used for FTIR retrieval fit |
|---|---|---|---|---|
| $NO_2$ | $(1.25\pm0.05)\times10^{-18}$ | 2830-2950 | HITRAN database | 2821-2859 cm⁻¹ |
| $CH_2O$ | $(1.27\pm 0.1)\times10^{-18}$ | 2600-2844 | Home-made calibration with (Gratien et al., 2007) | 2500-3000 cm⁻¹ |
| Trans-HONO | $(10\pm1)\times10^{-18}$ | 1200-1300 | Synthetic spectrum (Barney et al., 2001) | 1190-1310 cm⁻¹ 1200-1300 cm⁻¹ |
| $H_2O$ vapor | $(2.67)\times10^{-17}$ | 1150-2150 | HITRAN database and home-made spectra for high water concentration | 1190-1310 cm⁻¹ 1200-1300 cm⁻¹ |


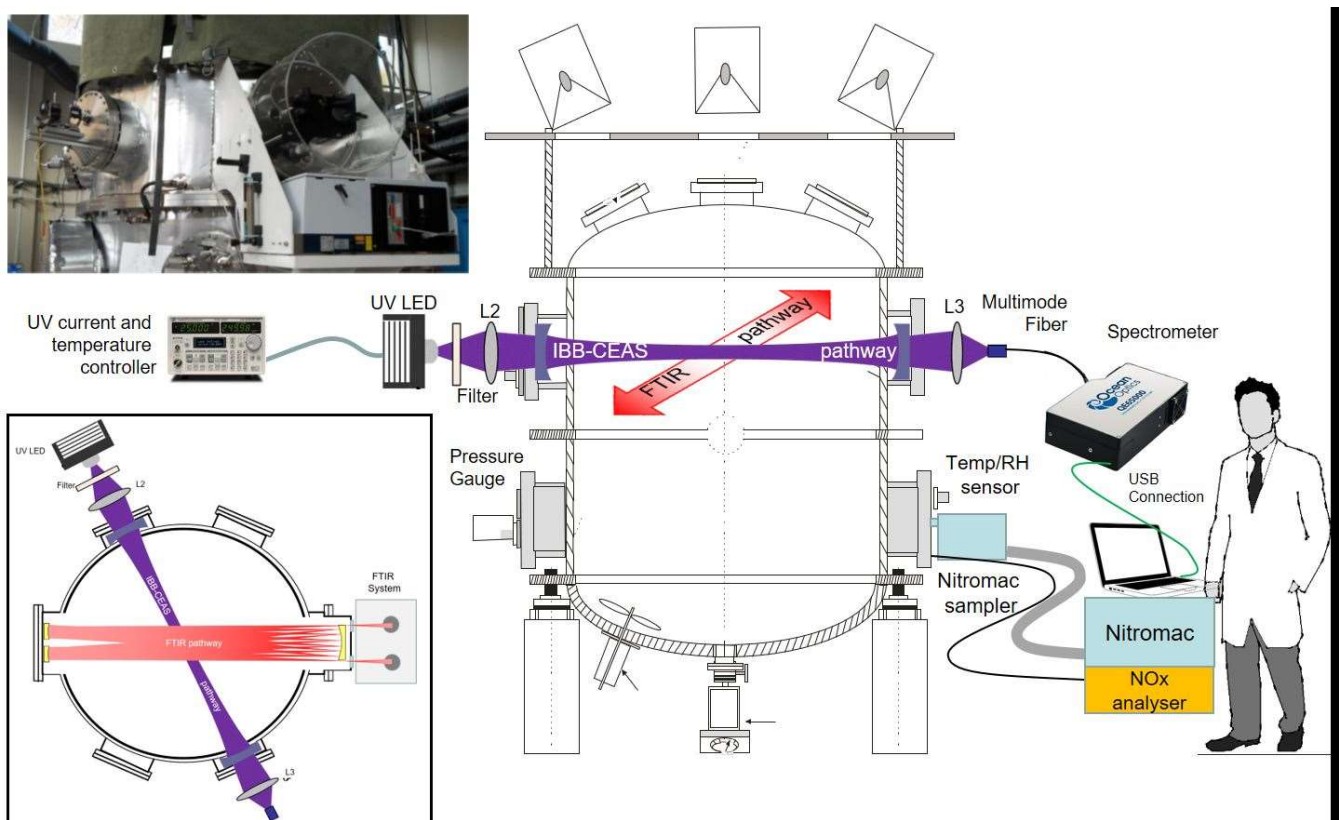

Figure 1 Schematics of the experimental set-ups and devices installed around the CESAM chamber for intercomparison: IBBCEAS, NitroMAC, FT-IR spectrometer, NOx analyzer, temperature and relative humidity sensor (T-RH sensor), pressure gauge. Insert display a photograph of the set-up (left-top) and a schematic view from the top showing the angle between the two in-situ spectrometric pathways (left-bottom). L2 and L3 are BK7 achromatic focus lens. Cavity mirrors had 25 mm in diameter, 2 m radius of curvature and 6.35 mm thickness.

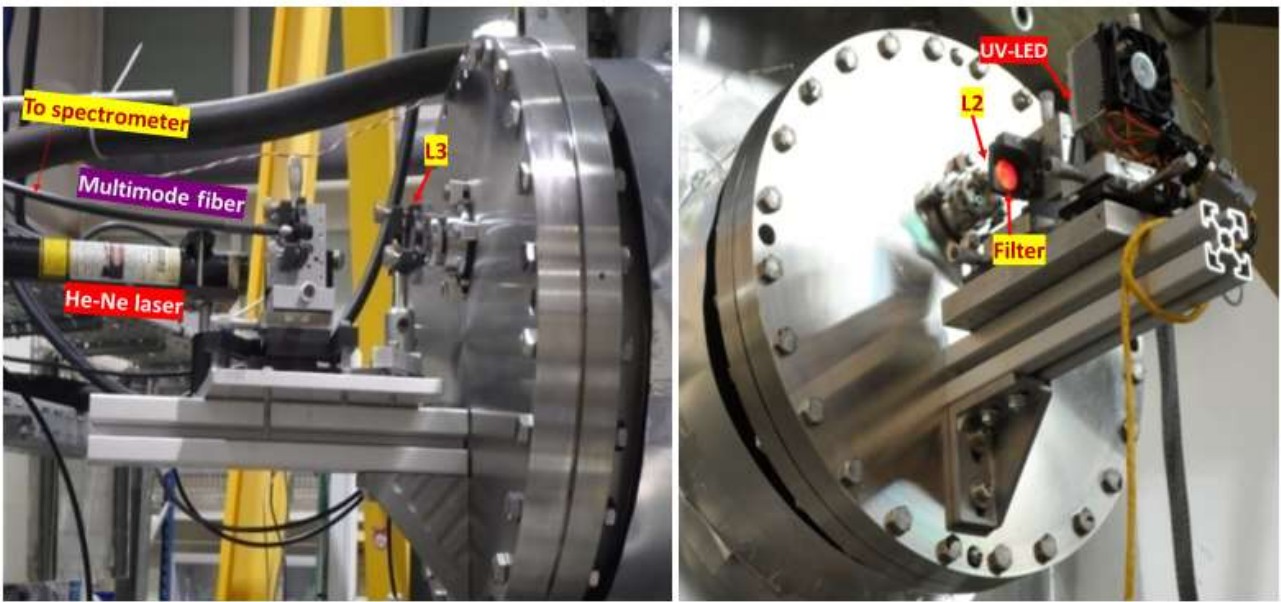

Figure 2 View of the IBBCEAS installation on the CESAM simulation chamber flanges. L2 and L3 are BK7 focus lens. M1 and M2 are two concave high-reflectivity mirrors.


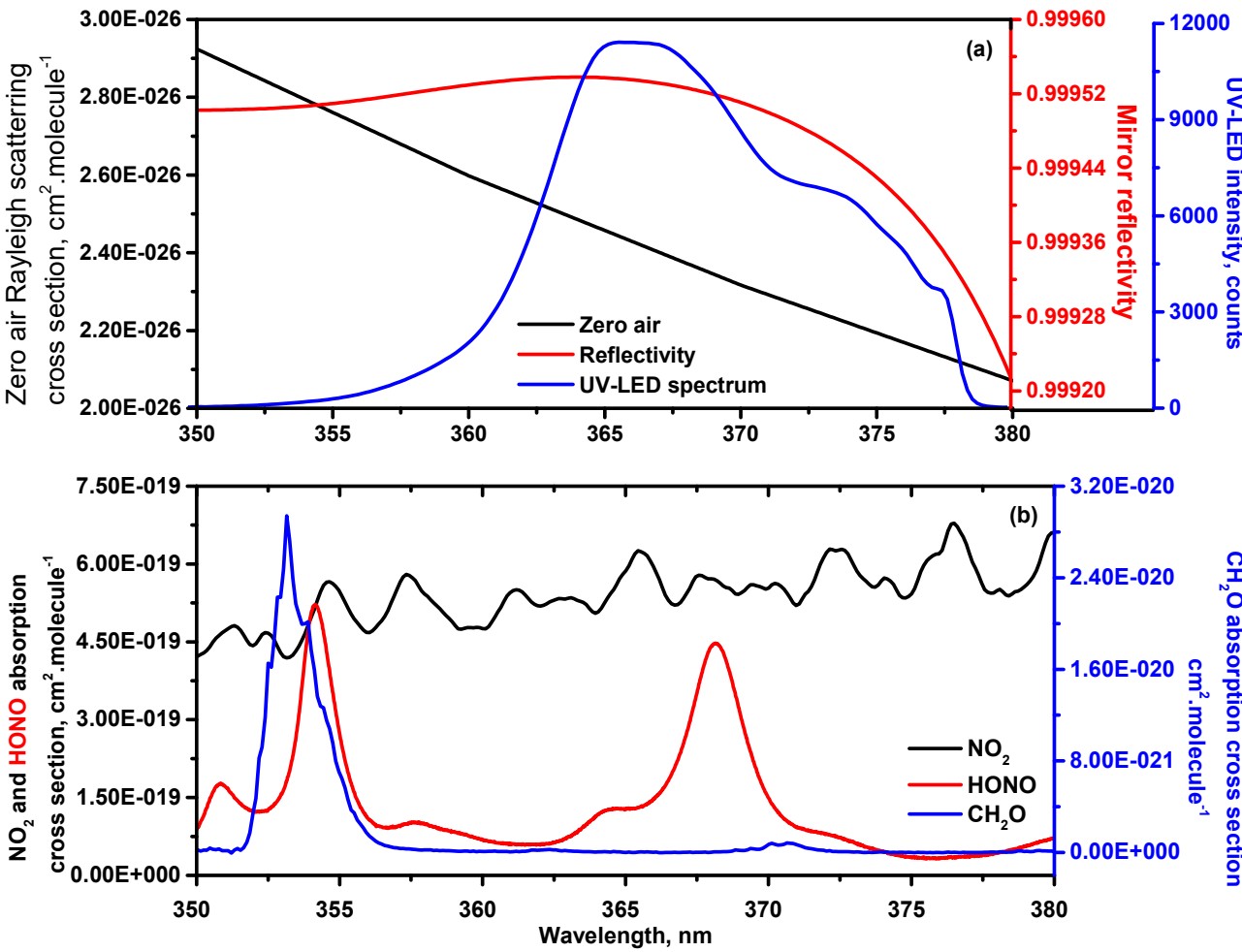

Figure 3 Spectral region of 350-380 nm for IBBCEAS measurements: (a) UV-LED emission spectrum (blue), wavelength-dependent mirror reflectivity (red) and Rayleigh scattering cross section of zero air (black) (Miles et al., 2001); (b) $NO_2$ (black), HONO (red) and $CH_2O$ (blue) absorption cross sections.

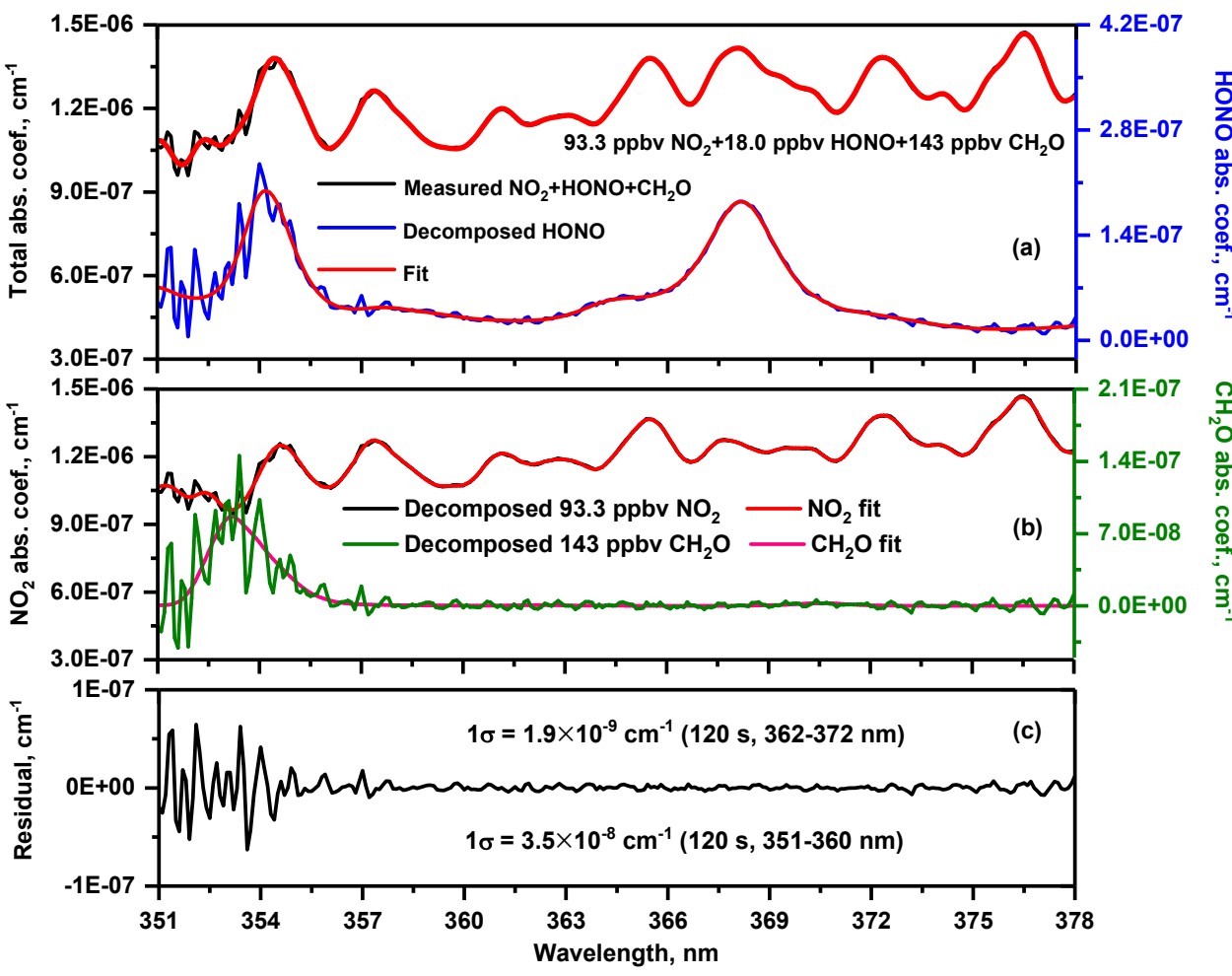


Figure 4 Measured and fitted NO₂, HONO and CH₂O spectra associated with the related residual. Decomposed spectra were specified in Ref. (Kennedy et al., 2011)

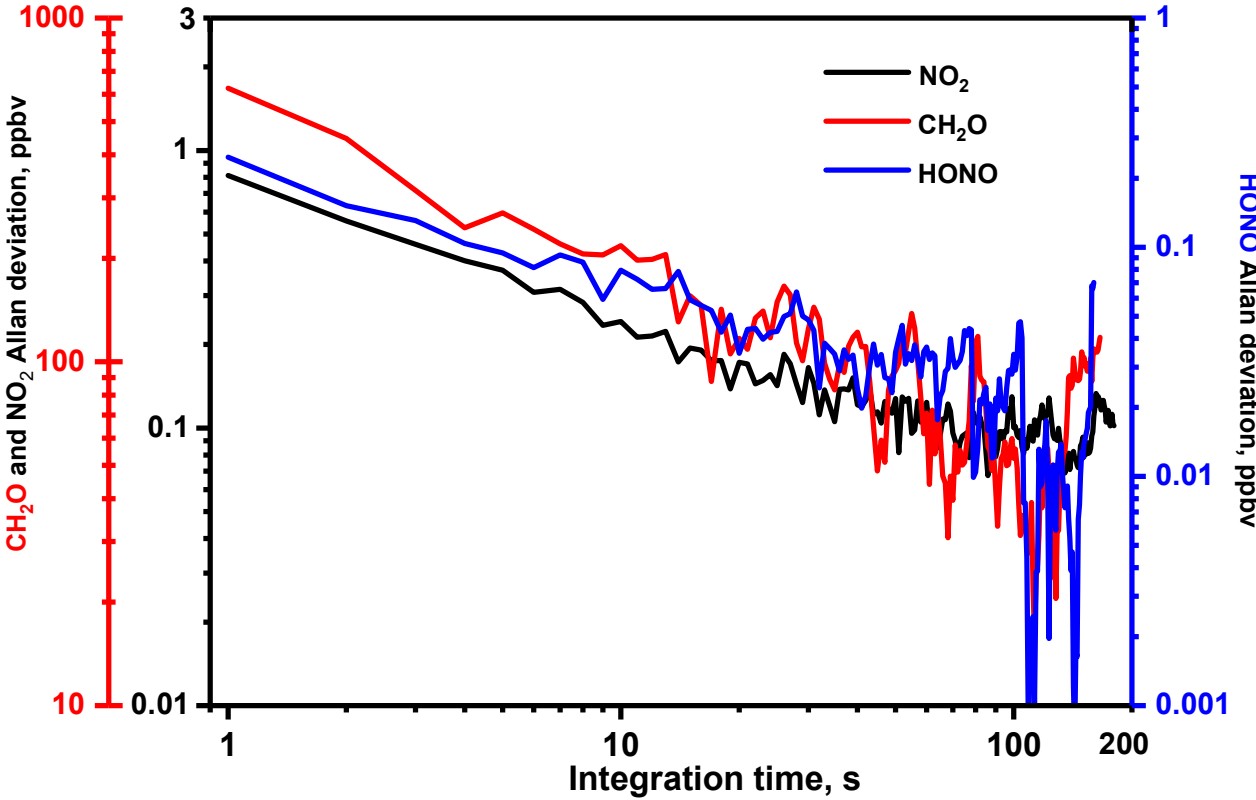


Figure 5 Allan deviation analysis for UV-LED IBBCEAS performance evaluation.

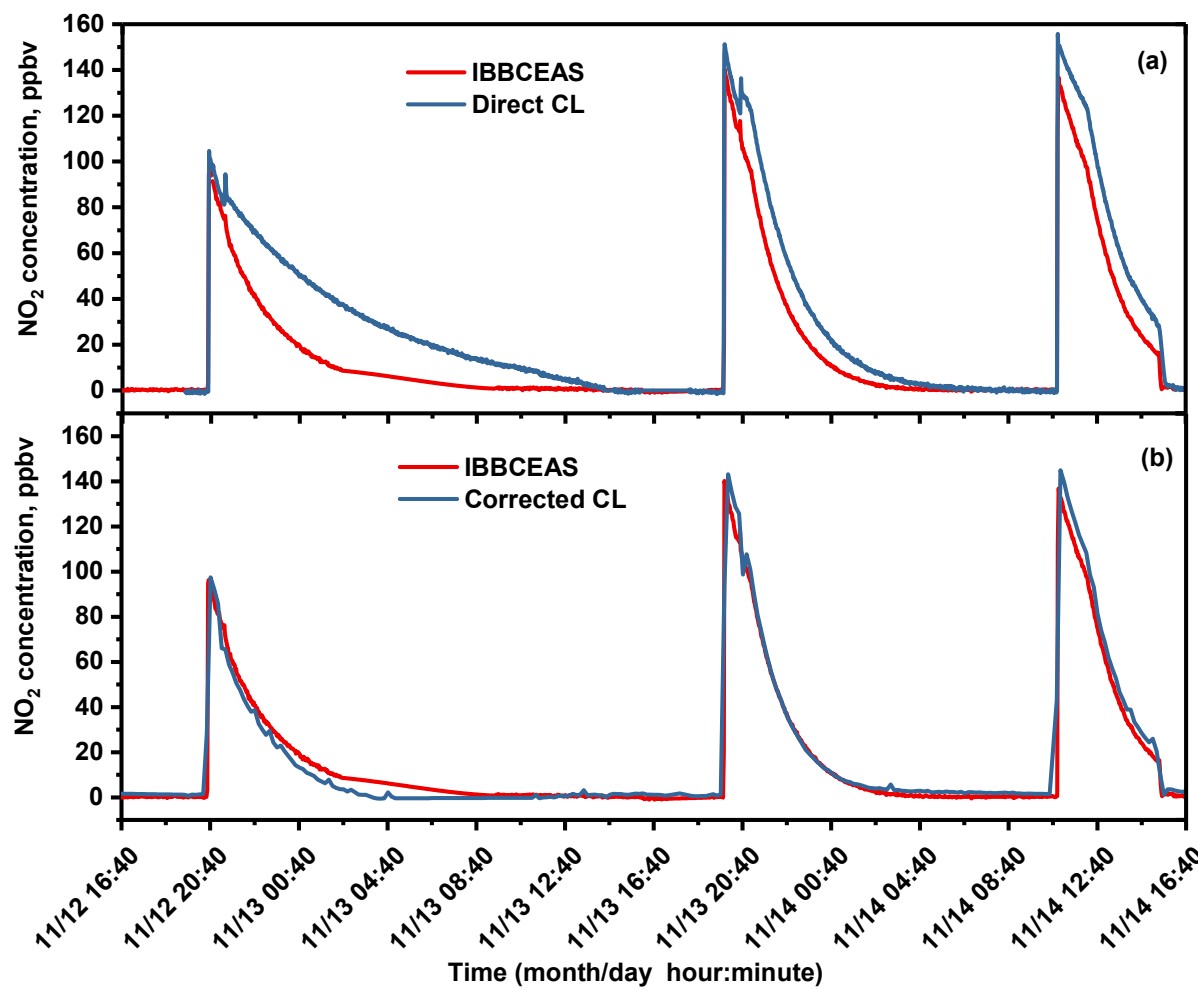

Figure 6 Investigation of positive interferences from nitrogen containing species (here: HONO) in NOx analyzer (CL) measurement, in comparison with UV-LED-IBBCEAS measurement: (a) from CL-NOx analyzer without HONO correction; (b) CL-NOx analyzer results after HONO correction.

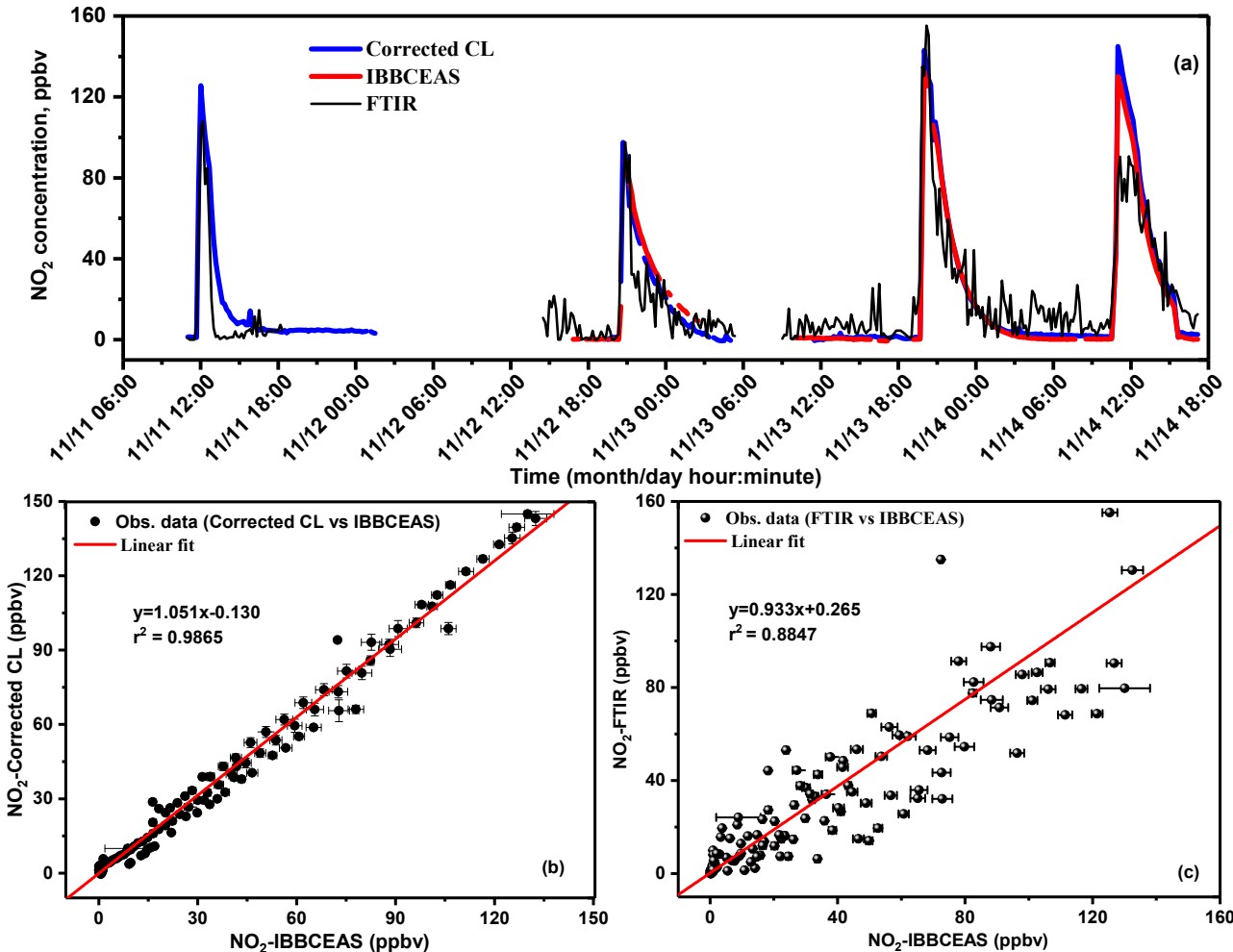

Figure 7 (a) Intercomparison measurement for NO₂ detection between IBBCEAS, FTIR and NOx analyzer after HONO interference correction; (b) Correlation of the measured NO₂ concentrations between UV-LED-IBBCEAS and NOx analyzer (CL) with HONO interferences correction; (c) Correlation of the measured NO₂ concentrations between UV-LED-IBBCEAS and FTIR.

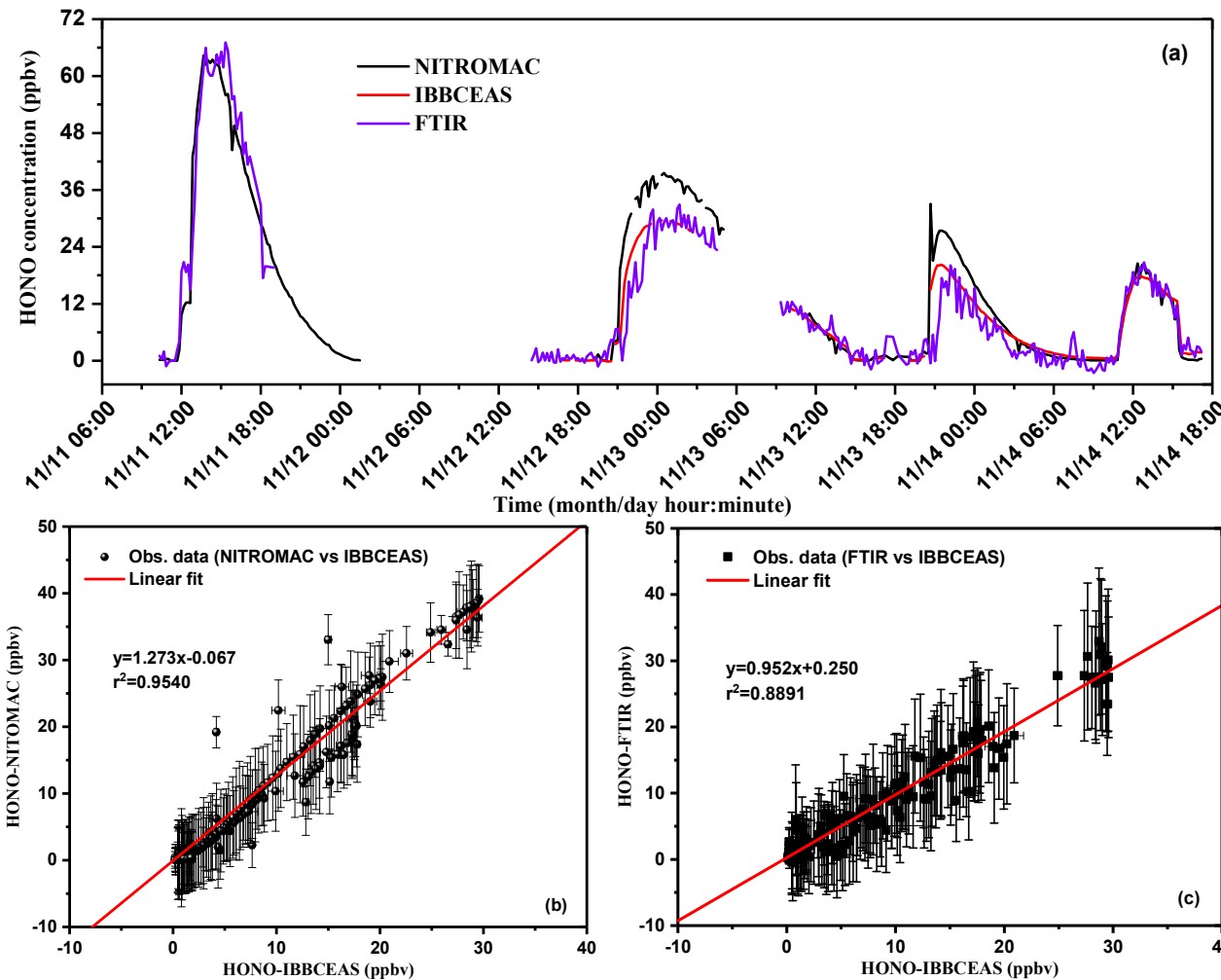

Figure 8 (a): HONO intercomparison measurements between IBBCEAS, NitroMAC and FTIR; (b): Regressions analysis for the correlation of the measured HONO concentrations using UV-LED-IBBCEAS and NitroMAC; (c): Correlation of the measured HONO concentrations between UV-LED-IBBCEAS and FTIR.

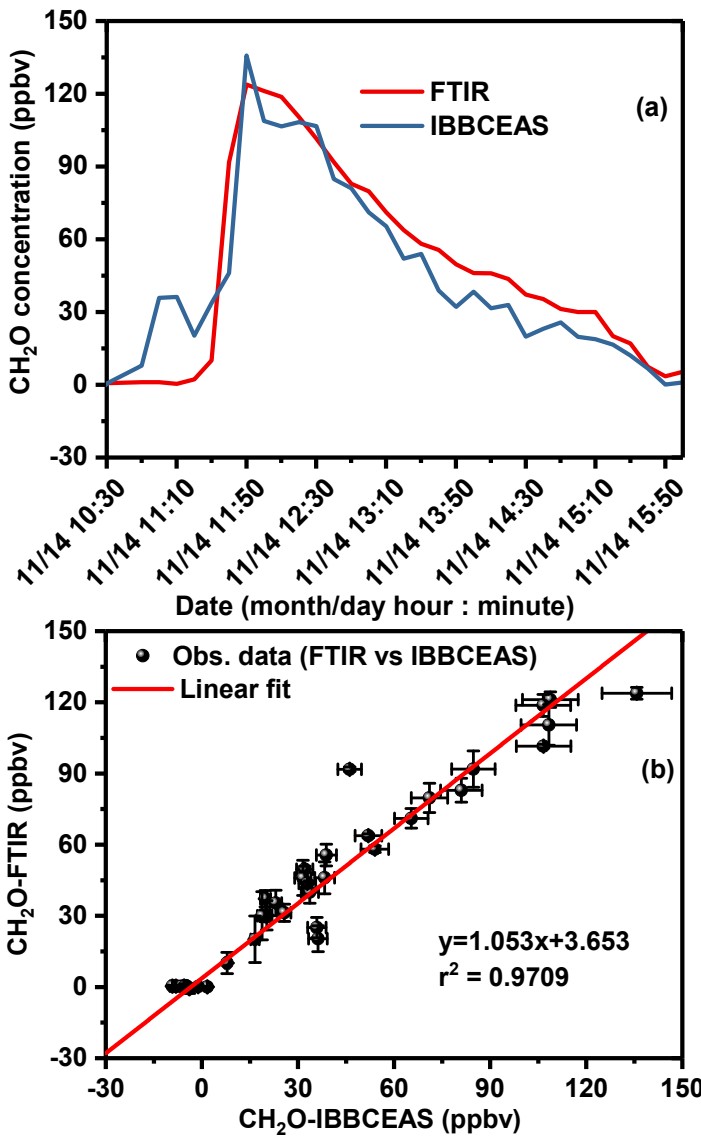

Figure 9 Intercomparison measurements of $CH_2O$ between IBBCEAS and FTIR: (a) Time series measurements of $CH_2O$ concentrations from UV-LED IBBCEAS and FTIR; (b) Linear regression of the measured $CH_2O$ in Fig. 9(a) : IBBCEAS (*x*-axis) versus FTIR (*y*-axis).


