# Peer review of "Intercomparison of IBBCEAS, NitroMAC and FTIR for HONO, NO2 and CH2O measurements during the reaction of NO2 with H2O vapor in the atmospheric simulation chamber of CESAM"

_Atmospheric Measurement Techniques, 2021_

## Referee Comment (RC1)

Review of "Intercomparison of IBBCEAS, NitroMAC and FTIR for HONO, NO$_2$ and HCHO measurements during the reaction of NO$_2$ with H$_2$O vapor in the atmospheric simulation chamber of CESAM" by H. Yi et al.

The article describe simultaneous measurements of NO$_2$ and HONO generated in a simulation chamber experiments by several in-situ monitors with a focus on UV LED based incoherent broadband cavity-enhanced absorption spectrometer (IBBCEAS) and its comparison against FTIR and chemiluminescent instruments. The experiments are well planned, and meticulous description of the instrumentation, measurements, intercomparisons and error analyses are provided. Well written, in a general sense, although small text corrections are sought. However, several aspects of the study need to be addressed before the paper can be accepted for publication. I suggest that the manuscript be majorly revised as per the comments listed below:

1. The diameter of the chamber is 2.13 m (page 3, line 88). Is it exactly the separation of cavity mirrors? Also, mirror radius of curvature is 2 m. Does radius less than mirror separation have any impact on cavity transmission or its stability?
2. In section 2.3.1, it is mentioned that $I_0$ is measured with nitrogen or dry (zero) air in the cavity. In the equation (1) for extinction coefficient, terms for Rayleigh scattering and Mie scattering are included. I think your target samples are trace quantities of NO$_2$, HONO, H$_2$O (vapor) and HCHO; all in gaseous state. When your $I_0$ already has zero air how can there be a contribution from Rayleigh scattering of air and Mie scattering by particles just by introduction of calibration gases? If your $I_0$ was measured in vacuum, then Rayleigh scattering by gases would be significant. I could not find any introduction of aerosol into your cavity either. Therefore, the equation (1) is misleading; also true for the equation (2) for mirror reflectivity retrieval where second term inside the parenthesis cannot be detected with dry air present in the $I_0$ spectrum.
3. In fact, if one looks at figure 3 it can be seen that the Rayleigh scattering cross section is 6 − 7 orders of magnitude smaller than the absorption cross sections of the calibration and target gases. Therefore, including zero air scattering cross section spectrum in the calculations makes no sense to me. I suggest to correct section 2.3.1 appropriately.
4. On page 11, line 212, a singular value decomposition method seems to have been adopted for analyzing data. No elaboration or specifics about this method could be found in the article. It is suggested that the method is elaborated in detail or appropriate reference(s) be specified.
5. How did you calculate the overall uncertainty from the individual uncertainties that you specified?
6. Line 220 on page 8: It is mentioned that the statistical uncertainty from the fit was included in your error analysis. No mention was found on how this was calculated and what is the magnitude you obtained as uncertainty from the fit. How did you estimate this?
7. From figure 3 (b) it looks like HONO has two absorption peaks, the first one close to 355 is interfering with an absorption peak of HCHO. In fact, after 357 nm, there is no influence (spectral interference) of HCHO presence on NO$_2$ and HONO. Interestingly, the LED has an order of magnitude less power below 357 nm. If it is just the NO$_2$ and HONO measurements that the authors are interested in, use the spectral region from 360 nm would do the job. From figure 4 it is amply clear that in the 366 − 372 nm region, HONO and NO$_2$ absorption can be clearly distinguished, then what is the need to extend it to 351 nm where light intensity itself goes negligible?

8.  While calculating the uncertainty, were the choice of spectral (sub) intervals used for analyzing different species considered? For example, like I said in the above comment, full spectral band or a sub interval of 360 – 378 can be used to analyze for $NO_2$ and HONO. Would the results be the same? Could you get the same uncertainties for both spectral intervals? If different, then the error analysis must include the errors due to the choice of spectral interval. In your case, if HCHO is the target species, then analyzing it in the 351 – 360 nm band may be desirable than the full window (of course, with $NO_2$ and HONO as co-analytes). If there is an effect due to spectral window choice, then optimization of the spectral interval of analysis for better results is desired.

9.  In the section 2.3.2 on page 8, FTIR spectral analysis is described. Spectral fit in support of FTIR measurements, like corresponding to figures 3 and 4 for IBBCEAS instrument, needs to be shown.

10. In this study authors used "synthetic" reference spectrum by Barney et al., 2001, for analyzing HONO from FTIR measurements. On page 10, line 303, the authors mention that the use of a different cross section for HONO introduced a larger error. Could you do an error analysis for FTIR measurements similar to that performed for IBBCEAS as well and add to the paper?

11. On page 9, line 274, comparison between IBBCEAS and FTIR measurements is specified. Although the discrepancy between the measurements is attributed to lower detection sensitivity of FTIR, the measurements of concentrations at well above detection limits show considerable deviations (Figure 7c). So, this is not the reason as mentioned in the paper. Either the FTIR is underestimating the $NO_2$ concentrations due to some bias or the IBBCEAS is overestimating. Spectral fit from FTIR would help in this situation to assess the spectral validation of the measurements. I would tend to believe IBBCEAS measurements, more so because its calibrations and spectral validations are provided whereas less information regarding FTIR spectra and fit are provided in the paper. Error bars are given for IBBCEAS side, but no uncertainty analysis seems to be available for FTIR. This needs to be addressed better than attributing deviations to lower sensitivity of FTIR. Line 339-340 in the section 4 (Conclusions) need to be corrected accordingly.

12.  This paper has both FTIR and IBBCEAS mounted on the wall of an atmospheric simulation chamber with two separate pieces (like a transmitter-receiver arrangement or the like) with atmosphere in-between. This is done to directly use atmosphere as sampling volume and avoid sampling losses in extracting it into a containing volume. This is a step towards applying this technique into the real atmosphere. However, the real atmospheric conditions may have other interfering gases and suspended aerosol (absorbing, non-absorbing, SOA, etc.). Many atmospheric simulation chamber experiments assessing such measurement methodologies specify the instrument performances under varying atmospheric conditions. How would your measurement sensitivity be affected in the presence of aerosols in the chamber?

13. Finally, a few suggestions for text edits:
    a.  On page 2, line 43, "topics" to be replaced by "topic"
    b.  Line 52, "following" to be replaced by "followed"
    c.  On page 6, line 163. The sentence starting with "When $NO_2$ concentration …." is incomplete.
    d.  Line 168. The sentence starting with "As described …" is incomplete.

---

## Referee Comment (RC3)

**Intercomparison of IBBCEAS, NitroMAC and FTIR for HONO, NO2 and HCHO measurements during the reaction of NO2 with H2O vapor in the atmospheric simulation chamber of CESAM**

Hongming Yi1\*, Mathieu Cazaunau2, Aline Gratien2, Vincent Michoud2, Edouard Pangui2, Jean-5 Francois Doussin2, Weidong Chen1

1Laboratoire de Physicochimie de l'Atmosphère, Université du Littoral Côté d'Opale, 59140 Dunkerque, France 2Laboratoire Interuniversitaire des Systèmes Atmosphériques, CNRS UMR7583, Universités Paris-Est-Créteil et Université de Paris Diderot, 94010 Créteil, France

\*now with Department of Civil and Environmental Engineering, Princeton University, Princeton, NJ 08544, USA

10 Correspondence to: Jean-Francois Doussin (Jean-Francois.Doussin@lisa.u-pec.fr) and Weidong Chen (chen@univ-littoral.fr)

Abstract. We report on applications of ultraviolet light emitted diode based incoherent broadband cavity enhanced absorption spectroscopy (UV-LED-IBBCEAS) technique for optical monitoring of HONO, NO2 and CH2O in a simulation chamber. Performance intercomparison of the UV-LED-IBBCEAS with a wet chemistry-based NitroMAC sensor and a

- 15 FTIR spectrometer has been carried out on real time simultaneous measurement of HONO, NO2 and CH2O concentrations during the reaction of NO2 with H2O vapor in the CESAM atmospheric simulation chamber. 1-σ (SNR=1) detection limits of 200 pptv for NO2, 100 pptv for HONO and 5 ppbv for CH2O over 120 s were found for the UV-LED-IBBCEAS measurement. On the contrary to many set-ups where cavities are installed outside the simulation chamber, we describe here an original in-situ permanent installation. The intercomparison results demonstrate that IBBCEAS is a very well suitable
- 20 technique for in situ simultaneous measurements of multiple chemically reactive species with high sensitivity and high precision even if the absorption bands of these species are overlapped. It offers excellent capacity to non-invasive optical monitoring of chemical reaction without any perturbation. For the application to simulation chamber, it has the advantage to provide a spatially integrated measurement across the reactor and hence to avoid point sampling related artefact.

**1** Introduction**

25 Atmospheric nitrous acid (HONO) is known as a major source of hydroxyl radicals (OH) (Harris et al., 1982; Finlayson-Pitts et al., 2000) in the atmosphere through its photolysis:

**$HONO + hv (**

sources include heterogeneous reactions, homogeneous gas-phase reactions, direct emission, surface photolysis, and biological processes, respectively (Spataro et al., 2014). HONO formation through one mostly possible heterogeneous reaction of NO2 with water (H2O) on surfaces is as follows:

 $2 \text{ NO}_2 + \text{H}_2\text{O} \rightarrow \text{HONO} + \text{HNO}_3$

(R2)

(R3)

35 HONO can be also formed through homogeneous chemistry with the following reaction:

**$\mathrm{NO} + \mathrm{OH} + \mathrm{M} \rightarrow \mathrm{HONO} + \mathrm{M}$**

[revised manuscript text omitted]
 NO2 cylinder: Air LiquideTM, AlphagazTM 99.9% purity) leading to about 120 ppbv of NO2 in the CESAM chamber. When NO2 concentration inside the chamber stabilized at 120±5 ppbv, water vapor produced in a small pressurize stainless steel vessel filled with ultrapure water (18.2 Mohm, ELGA Maxima). The relative

humidity (RH) inside the chamber was allowed to increase to ~60% at 23 °C (corresponding to an absolute H2O vapor mixing ratio of ~1.6%).

Under these conditions, as described by Wang et al. (2011), a significant gas-phase HONO is systematically observed. As described in the literature (Finlayson-Pitts et al., 2000; Lammel et al., 1995; Spataro et al., 2014). It is generated through heterogeneous formation on chamber inner surfaces via a complex reaction of NO2 with adsorbed  $H_2O$  chamber walls. All

170 the instruments (UV-IBBCEAS, FTIR, NitroMAC, NOx analyzer, temperature and humidity sensor, pressure gauge) simultaneously recorded the relevant data (including NO2, HONO, NO and H2O concentrations, temperature and pressure) for data analysis and instrument intercomparison. Absolute NO2 concentrations obtained by FTIR were used for the determination of cavity mirror reflectivity.

Four NO2 injections in the presence of humid air were organized during the four days of experiments. During the last

- 175 experiment, an injection of formaldehyde (HCHO) was performed to allows the investigation of the sensitivity of the UV-IBBCEAS data analysis to the interferences in the UV range. Formaldehyde was prepared by sublimating commercial paraformaldehyde (CH2O)n (Fluka, "extra pure" grade) under vacuum in a glass line and collected at a known pressure in a bulb of known volume. This quantity was then flushed into the chamber with a gentle flow of pure nitrogen. A controlled dilution flow was allowed to the chamber to induce a forced decrease of the sampled concentrations and hence testing the
- 180 quantification performance of the various analytical devices across a few orders of magnitudes. This process explains the peak shape formed of a straight injection step followed by an exponential decay of the various experiences.

**2.3 Data analysis**

**2.3.1 UV-LED-IBBCEAS**

In an IBBCEAS experiment, the transmitted spectra  $I_0(\lambda)$  from the cavity without absorbing species are firstly measured by 185 filling the cavity with pure N2 or zero air, and then the spectra  $I(\lambda)$  in the presence of target sample are recorded. The absorption by molecular species, Rayleigh scattering by gas mixture  $\alpha_{Ray}(\lambda)$ , Mie scattering by particles  $\alpha_{Mie}(\lambda)$  and absorption by particle  $\alpha_{abs-particle}(\lambda)$  contribute to optical light extinction in the cavity, the total optical extinction coefficient  $\alpha(\lambda)$  ean be given as below. (Yi et al., 2016):

$$\alpha(\lambda) = \left(\frac{1 - R(\lambda)}{d} + \alpha_{Ray}(\lambda) + \alpha_{Mie}(\lambda) + \alpha_{abs-particle}(\lambda)\right) \times \left(\frac{I_0(\lambda)}{I(\lambda)} - 1\right)$$
(1)

190 where d is the distance between two cavity mirrors.

In the present work of gas-phase chemical reaction in the simulation chamber filled by zero air, the chamber is free of particles, thus  $a_{Mie}(\lambda)\approx 0$  and  $\alpha_{abs-particle}(\lambda)\approx 0$ . The mirror reflectivity can be determined by using a known-concentration NO2 sample:

$$R(\lambda) = 1 - d \left( \alpha_{NO_2} \times \frac{I_{NO_2}(\lambda)}{I_{Zero \ air}(\lambda) - I_{NO_2}(\lambda)} - \alpha_{zero \ air}^{Ray} \right)$$
(2)

- 195 where  $I_{\text{NO2}}$  and  $I_{\text{zero air}}$  are the transmitted LED light intensities through the cavity containing NO2 and zero air, respectively,  $\alpha_{\text{NO2}}$  is the absorption coefficient of NO2 and  $\alpha_{\text{Ray-
[revised manuscript text omitted]

---

## Author Comment (AC4)

**RC2**: 'Comment on amt-2021-19', Anonymous Referee #2, 12 Apr 2021

The paper describes a IBBCEAS which, on the contrary to many existing set-ups, introduces the innovation of its in-situ installation, avoiding unwanted invasive use of pumps, etc. The system can measure HONO and, simultaneously, $NO_2$ and $CH_2O$. To evaluate its performance, an intercomparison against other instruments, NitroMAC, FTIR and NOx monitor is carried out. The paper is well written and results are well discussed. There is a detailed description of the instrumentation, procedures and error analysis. For these reasons I recommend its publication after considering the following aspects:

24: The title says HONO, $NO_2$ and $CH_2O$, but the introduction mainly talks about HONO. HONO measurement is challenging, while the detection of $NO_2$ and $CH_2O$ is better stablished. Nevertheless, I would suggest to either include brief information on $NO_2$ and $CH_2O$ or explain that the main interest is measuring HONO although $NO_2$ and $CH_2O$ absorb in the same region and are also tracked, being an advantage of the technique.

**Response** : We agree with the reviewer's opinion, the following sentences have been added in the introduction section (page 3, lines 69-72):

"Although the main interest for current work is to measure HONO, $NO_2$ and $CH_2O$ are two other important atmospheric species (Washenfelder et al., 2016; Liu et al., 2020), these two molecules have strong absorption in the same region. Simultaneous measurements and quantification of HONO, $NO_2$ and $CH_2O$ can be performed by the IBBCEAS techniques (Wu et al., 2014; Washenfelder et al., 2016; Duan et al., 2018; Jordan et al., 2020)."

80: This work introduces some changes in the set-up of the instrument, but it is based in previously developed IBBCEAS. Please, add some references.

**Response** : Four references related to previously reported IBBCEAS (Gherman et al., 2008; Fuchs et al., 2010; Wu et al., 2012; Wu et al., 2014; Duan et al., 2018; Jordan et al., 2020) have been added into section 2.3.1 of the revised manuscript (page 7, line 198-199):

There are only two references related to measurements of $NO_2$ and HONO in simulation chamber (Gherman et al., 2008; Fuchs et al., 2010). Two recently published papers reporting on measurements of NO2 and HONO in ambient air have been added as references:

Duan, J., Qin, M., Ouyang, B., Fang, W., Li, X., Lu, K., Tang, K., Liang, S., Meng, F., Hu, Z., Xie, P., Liu, W., and Häsler, R.: Development of an incoherent broadband cavity-enhanced absorption spectrometer for in situ measurements of HONO and $NO_2$, Atmos. Meas. Tech., 11, 4531–4543, doi:10.5194/amt-11-4531-2018, 2018.
Jordan, N. and Osthoff, H. D.: Quantification of nitrous acid (HONO) and nitrogen dioxide ($NO_2$) in ambient air by broadband cavity-enhanced absorption spectroscopy (IBBCEAS) between 361 and 388 nm, Atmos. Meas. Tech., 13, 273–285, doi:10.5194/amt-13-273-2020, 2020.

225: Can you confirm that DL for $CH_2O$ is 5 ppb? The emission of the LED below 356 nm is very low (Fig 3). The absorption for 143 ppb in Fig 4 doesn't seem to suggest that an absorption of 5 ppb will be detectable with such noise. DL has been calculated from 1-σ in Fig 4 through the region 351-378 nm as it is the analysis

region, but 1-σ in the region where $CH_2O$ absorbs is much higher, therefore, the real DL would be higher. That noise would also explain the noisy profile in Fig 9. Please, comment.

**Response** : Spectral region of 351-378 nm was used to fit, when we calculated 1σ minimum detectable concentration (MDC) for HONO and $NO_2$, we used 362-372 nm residual data. But for $CH_2O$, the spectral data of 351-360 nm was used to estimated 1σ MDC. MDC (or DL) for $CH_2O$ should be 41 ppbv not 5 ppbv with 120 s. We have corrected this error. The corresponding text in section 2.3.1 page 8, lines 242-244 has been thus revised as follows:

"Based on the fit residual, the corresponding 1σ minimum detectable concentration (MDC) with mixing ratio for 120 s integration time are 112 pptv for $NO_2$, 56 pptv for HONO using 362-372 nm region data. MDC for $CH_2O$ with 120 s is 41 ppbv by using of 351-360 nm spectral data."

580: There are -15 ppb of $CH_2O$ in Fig. 9. It might be due to interference with HONO. On the one hand, in general, these unrealistic data can be withdrawn as they are below the DL. Indeed, those data seem to have been withdrawn from Fig. 9b since, looking at the 0 ppb of concentration for IBBCEAS, data for NITROMAC do not replicate the whole set of data in Fig. 9a, so they can be removed from Fig. 9a. On the other hand, they give information on how HONO is interfering, therefore, if the authors decide to include these data, some comment should be made in the text.

**Response** : Based on our updated analysis, 1σ minimum detectable concentration (MDC) for $CH_2O$ is about 41 ppbv with 120 s, the data of about -15 ppbv of "$CH_2O$" in Fig. 9 before the introduction of $CH_2O$ sample (without $CH_2O$) are below the MDC, thus these unrealistic data before injection of $CH_2O$ have been withdrawn in the revised Figure 9a, as shown below. Because the measured $CH_2O$ concentration below 41 ppbv is not accurate. In the revised version, some unrealistic data have been withdrawn.

[Figure]

157: There were 4 experiments. At the beginning, the first experiment is described, and in line 174, it is said that there were 4 days of experiments. It can be mentioned that they were done under the same conditions as the first one.

**Response** : Yes, the 2[th] to 4[th] experiments were performed under the same experimental conditions as the first one. The procedures for 4 experiments were the same: firstly, the simulation chamber was pumped and evacuated to a pressure of ~ 1 mbar; secondly, the simulation chamber was filled with zero

air to 1 atm (1000 mbar); and then, $NO_2$ (<150 ppbv) sample was injected into the chamber for mirror reflectivity determination; finally, $H_2O$ vapor (<1.86%) were introduced into the chamber for HONO generation. During the whole experiment, IBBCEAS, NITROMAC, FTIR, NOx analyzer, temperature and relative humidity sensor (T&RH sensor), pressure sensor were running to record all related data for later analysis. The following description was added on page 6, lines 176-177 to describe the experiment procedure:

"There were 4 experiments during the whole measurement, the $2^{th}$ to $4^{th}$ experiments were performed under the same experimental conditions as the first one. The four experiments were followed by the same procedure."

172 and 198: Cavity mirror reflectivity is a key parameter in IBBCEAS for calculating the concentrations of the target molecules. Having a $NO_2$ monitor, why did you use FTIR for its determination? The NOx analyzer shouldn't have interferences during calibration as $NO_2$ pure is introduced and there is no NOy (unless RH was not zero in the chamber). Is it related to accuracy? Please, add some comment.

**Response** : Because the RH in the chamber was not ideally zero, residual $H_2O$ vapor always existed inside the chamber, the estimated residual $H_2O$ concentration is about 0.002% to 0.01% from T&RH sensor and FTIR. Once $NO_2$ was introduced into the chamber, unknown-concentration NOy would be generated immediately. As discussed in the later section, positive interferences can't be avoided. So FTIR spectrometer was used to determine $NO_2$ concentration for get more accurate mirror reflectivity in the present work.

240: Table 1 reflects the spectral regions corresponding to the IBI. Are these the analysis regions used for the analysis of each compound? If not exactly, please include this information in Table 1 or in the text.

**Response** : The table has been revised accordingly and the spectral windows used for the FTIR data analysis have been added.

255: Rephrase: "and 120 such acquisition data"… to "and 120 of such acquisition data" or "120 data acquired in this manner were"

**Response** : thanks reviewer for such helpful revision, correction has been done with "120 data acquired in this manner were".

276: detection limit of 10 ppbv at a sampling time of 1 min, compared …. (or similar, to distinguish from DL of 5 ppb at sampling time of 5 min in line 130).

**Response** : Agree with the reviewer, we corrected this typo error. The sentence is changed to "MDC of 10 ppbv at a sampling time of 5 min compared to that of 112 pptv in 2-min for IBBCEAS" (page 10, lines 301-302).

277: In Fig 7a, $NO_2$ by FTIR is underestimated when there is $CH_2O$. Was $CH_2O$ included as pure reference spectra in the analysis of $NO_2$?

**Response** : First of all, we thank the reviewer for having spotted this. Indeed, because of the presence of many strong water lines under the condition of the experiment in the 1500-1900 cm$^{-1}$ region, the spectral region used to estimate NO$_2$ concentration was 2830-2950 cm$^{-1}$. The choice of this region had two consequences:

- as the 2900 cm$^{-1}$ region is far from exhibiting similar intense NO$_2$ absorption, the random error from the NO$_2$ quantitation was brought to ca. 20 ppbv.

- while CH$_2$O was systematically included as pure reference spectra in the analysis leading to NO$_2$, there seem to be an interference between the two species during the fit. We double check this but it seems that this effect is rather due to some noise addition from the subtraction of the HCHO reference spectrum which is an experimental spectrum (and not a calculated one) and so which contain some noise.

This was not totally unexpected and this is why we relied on the chemiluminescence analyzer NO$_2$ data (if corrected from HONO interferences) which fit well with IBBCEAS measurements.

295: The reference from Stutz is widely used by the scientific community showing good agreement with others, but not with Brust. Apart of the error due to using different HONO references, the hypothesis of the mixing fans makes sense, and then I wonder: it would imply that also in the first peak in Fig 7 the mixing fan speed was increased since NitroMAC tracks FTIR data as in the 4$^{th}$ peak, while it doesn't in the 2-3$^{rd}$ peaks. How was the mixing fan in the day of the first peak?

**Response** : Unfortunately, at the time of these experiments there was no recording of the mixing fan speed and the current brought to the mixing system electric motor was manually adjusted. The lab notebook only reports a change in the setting in the morning of the 4th peak days.

This set of experiments has latter led the CESAM group to implement the recording of the fan mixing speed that is operational now (but too late to support the hypothesis made here).

300: "NitroMAC values were slightly larger than IBBCEAS". Slope NitroMAC vs IBBCEAS is 1.27, I would remove the word slightly.

**Response** : Agree with the reviewer, we have deleted the word "slightly"

325: "relative low detection limit" do you mean high instead of low?

**Response** : Agree with the reviewer, we have corrected "low" to "high".

334: The authors might comment on how feasible is to use this in-situ system in other chambers.

**Response** : The current approach can be easily extended to application in other chambers for in-situ optically "watching" chemical reaction without introduction of any disturbance. The challenge is how to keep high precision, enough temporal resolution, and good stability if the harsh experiment condition happens, such as aerosols or other suspended particles are introduced into the chamber.

575: Fig 8c, HONO IBBCEAS doesn't have error bars.

**Response** : HONO from IBBCEAS has error bars, but the magnitude of its error bar is much smaller than that of HONO from FTIR, because the measurement precision of IBBCEAS is about 100 times better than that of FTIR. BTW, three wrong error bars for HONO from IBBCEAS in Fig. 8b have been corrected as well.

575: Did the authors try to analyze HONO (and $NO_2$) in the region 357-380 nm? To analyze $CH_2O$, the selected region is adequate as both the HONO absorption at 368 nm and $NO_2$ in the whole region help in a better calculation of the HONO fit, and therefore, the error due to the interference of HONO around 353 nm when calculating HCHO is reduced. But, to analyze HONO and even $NO_2$, the absorption of HONO at 368 nm is high enough to determine it in 357-380 nm, and it would avoid the interference with $CH_2O$ in a region with high noise. Would Fig. 8(a) be the same?

**Response**: Based on the fitted residual (Fig. 4(c)), using the 362 – 372 nm spectral to analyze $NO_2$ and HONO, MDC for $NO_2$ and HONO will be improved to 112 pptv and 56 pptv, respectively. For $CH_2O$ measurement, 351-360 nm spectral range is used to retrieve $CH_2O$ concentration, the MDC for $CH_2O$ will be 41 ppbv with 120 s. Based on the analysis, the corresponding text in section 2.3.1 (page 8, lines 242-244) have been revised as follows:

"Based on the fit residual, the corresponding $1\sigma$ minimum detectable concentration (MDC) with mixing ratio for 120 s integration time are 112 pptv for $NO_2$, 56 pptv for HONO using 362-372 nm region data. MDC for $CH_2O$ with 120 s is 41 ppbv by using of 351-360 nm spectral data."

Some editing comments:

80 and 87: installed on à installed in
108: reagent as soon as a few à reagent a few?
115: standard solutions was à standard solutions were
126: (see Fig. 1-insert) à (see Fig. 1)
155: 370), a FTIR spectrometer à 370) and a FTIR spectrometer
157: The experiment, the experiments or the first experiment?
163: Check sentence "When…"
168: Check sentence "As described…"
175: allows à allow
268: between two instruments à between the two instruments
279: weighed à weighted
565: Figure 6, X axis. Month (not moth)

**Response** : All these editing errors have been corrected.

---

## Author Comment (AC5)

RC3: 'Comment on amt-2021-19', Anonymous Referee #3, 22 Apr 2021

**Reviewer 3** (particular thanks to the reviewer for the careful reading and revision)

This manuscript reports an open path setup established at the CESAM simulation chamber in Paris, which is based on Incoherent Broadband Cavity Enhanced Absorption Spectroscopy (IBBCEAS). This instrument for the detection of HONO, $NO_2$ and $CH_2O$ is compared with other experimental approaches "NitroMac" and more conventional chemiluminescence detection as well as FT-IR spectroscopy. The performance of the instruments is characterized, and aspects of the instruments' advantages and drawbacks is discussed on basis of measurements taken in the course of a 3-day campaign.
The content of the manuscript is quite appropriate for the special issue on Atmospheric Simulation Chamber Research, instrument intercomparisons should be of general interest to the respective community. The manuscript is however not particularly well written as far as the use of the English language is concerned. In many sentences it was not very clear what the authors were trying to say. This should be improved in the final version of the submission (see the attached file, where also more comments can be found for the benefit of the authors).

**Response** : Thanks very much for the reviewer's valuable suggestions, we have carefully checked the English usage.

Other shortcomings are: There is a lack of detail in some parts. E.g. the retrieval of data (from the NitroMac machine and FTIR spectrometer) could have been discussed somewhat better. The discussion of systematic errors and of errors in general are of interest to the community and could have been done in more detail and more quantitatively. The role of aerosol was not even mentioned in the discussion – it is important in the context of the data retrieval and limits of detection. There is certainly not enough reference made to the relevant literature. The citations appear to be incomplete. The manuscript exclusively describes technical aspects of the detection setups for HONO, $NO_2$ and $CH_2O$ and does not report or discuss any new atmospheric or gas phase processes in the context of HONO formation or destruction, in other words, the advancement of science is minimal, but this was obviously not the main objective of this work.

**Response** : We revised the whole manuscript according to the reviewer's comments one by one as follows.

1. Page 1, line 16, "1-σ" reads strange, should be 1\sigma.

**Response** : corrected to "1σ".

2. Page 2, line 32, "one mostly possible heterogeneous" English usage.

**Response** : corrected to "the most possible heterogeneous".

3. Page 2, line 51, "... are sensitive. What does "sensitive" in this context mean. This is merely qualitative."

**Response** : Please see the answer in next response.

4. Page 2, lines 51-53, "They generally rely on conversion of HONO into nitrite ion ($NO^{2-}$) following by absorbing dye conversion (Kleffmann et al., 2006) and may be susceptible to chemical interferences and sampling artifacts (Stutz et al., 2010)." English usage.

**Response**:the statement has been revised as follows (Page 2, lines 51-55): "In wet chemical methods, HONO is sampled on aqueous/humid surfaces and converted into a species suitable to be analyzed with conventional chemical analytical techniques such as ion chromatography (IC), fluorescence (FL), chemiluminescence (CL), long-path absorption photometer (LOPAP) or high-performance liquid chromatography (HPLC)" (Chen et al. 2013). These wet-chemical-based instruments often suffer from unquantified chemical interferences and sampling artifacts (Stutz et al., 2010)."

W. Chen, R. Maamary, X. Cui, T. Wu, E. Fertein, D. Dewaele, F. Cazier, Q. Zha, Z. Xu, T. Wang, Y. Wang, W. Zhang, X. Gao, W. Liu, F. Dong, "Photonic Sensing of Environmental Gaseous Nitrous Acid (HONO): Opportunities and Challenges," in *The Wonder of Nanotechnology: Quantum Optoelectronic Devices and Applications*, M. Razeghi. L. Esaki, and K. von Klitzing, **Eds**., SPIE Press (ISBN 9780819495969), Bellingham, WA, pp. 693-737 (2013)

5. Page 2, lines 62-64, "frequently, intercomparison between in point and long-path measurements exhibited significant discrepancies with uncertainties of 10%-25% for HONO concentrations from ten-pptv to ten-ppbv range." This is too unspecific.

**Response**:Some description is not accurate and lack of results citation, the sentence has been corrected to (Page 3, lines 65-66) "frequently, intercomparison between in point and long-path measurements exhibited significant discrepancies with uncertainties of about 20% (Pinto et al., 2014; Kleffmann et al., 2006) in HONO concentrations varying from ten-pptv to ten-ppbv range".

6. Page 3, lines 69-70, "instrument for simultaneous measurement of wider concentrations at natural conditions of HONO (100 pptv-30 ppbv), $NO_2$ (100 pptv-120 ppbv) and $CH_2O$ (3-150 ppbv)". What motivates the ranges stated here. Seems in contradiction with the abstract.

**Response**:it is now stated as follows in order to avoid any confusion (like "generation of their concentrations in a wide range"): "evolution of concentrations of HONO (0-30 ppbv), $NO_2$ (0-120 ppbv) and $CH_2O$ (0-150 ppbv) in a wide range has been optically tracked using a single IBBCEAS device in the present work.

Abstract, page 1, lines 16-18: "$1\sigma$ (SNR=1) detection limits of 112 pptv for $NO_2$, 56 pptv for HONO and 41 ppbv for $CH_2O$ in 120 s were found for the UV-LED-IBBCEAS measurement approach".

Page 3, lines 74-75 : "instrument for simultaneous measurement of HONO (0 pptv-30 ppbv), $NO_2$ (0 pptv-120 ppbv) and $CH_2O$ (0-150 ppbv)."

7. Page 3, line 83, "plate". What is the divergence of the LED. What optical power was used for the experiment? What is the shape and spectral with of the LED's spectrum? This is not mentioned here but it is in Figure 3 it seems. The spectrum looks like it is affected by a filter - it that correct?

**Response**:"plate" was corrected to "block".

LED emission area was about 1 mm$^2$ with a divergence angle of about ±60$^\circ$, according to the datasheet. LED emission power was about 300 mW. The distance between LED and L2 (with diameter $d$=25.4 mm) was about 100 mm, which leads to an estimation of the used power of about 66 mW in the experiment. The shape of the LED emission spectrum was as shown in Fig. 3(a), experimentally recorded in the present work. The measured LED spectrum shapes were the same with and without filter, not affected by filter.

8. Page 3, lines 88-89, "on the simulation chamber walls". I guess they were on mirror holders and not on the walls.

**Response** : the cavity mirrors were indeed installed directly on the flanges attached to the simulation chamber walls, as shown in Fig. 2. On the mirror holders are installed the used focusing lens (L2) and the optical filter.

9. Page 3, line 92, These are effective pathlengths.

**Response** : yes, they are effective pathlengths. "equivalent" is replaced by "effective".

10. Page 3, line 93, L1 not shown in Figure 1. It is called L2. Over 30 nm one may want to consider a doublet (achromatic lens). How far from the LED was the lens placed? This distance is crucial. If the light was focussed (as stated), then what was the magnification of the first image in the cavity? Note, the authors have not mentioned the divergence of the LED and hence it is not clear how much optical power could actually be used for the IBBCEAS experiment. Where was the focal point.

**Response** : Yes, you are right (thanks for the careful review): L1 should be L2 (corrected). In reality, L2 and L3 were all indeed achromatic lens (now mentioned clearly in the revised manuscript). The distance between L2 ($f$=75 mm) and LED was adjustable (in the range of 80-100 mm) such that the focal point was located near the center of the optical cavity. The magnification of the first image in the cavity is about 10 times.

Answers to the questions about LED are provided above in Question 7.

11. Page 3, line 94, What is the suppression of the filter at 390 nm (and at 340 nm)? From Figure 3 it is not clear that these band-pass filter specs are appropriate since the reflectivity spectrum is only shown up to 380 nm.

**Response** : A 340-390 nm filter lets pass the used LED emission (350-380 nm, Fig. 3(a)) and cuts off the undesirable wavelengths ($\lambda$ < 340nm or $\lambda$ > 390 nm) out of the high-reflectivity range of the cavity mirrors in order to avoid CCD saturation due to the lower mirror reflectivity in these wavelength regions.

12. Page 4, line 96, "L2 (BK7, f=75 mm)". Contradicts the labelling in Figure 1.

**Response** :  right, it should be L3, corrected.

13. Page 4, line 97, Was the spectrometer temperature controlled? Was it cooled at all? Temperature stabilization is important to avoid wavelength drifts.

**Response** : The temperature of the used CCD spectrometer was controlled by a thermoelectric cooler (TEC) and cooled down to 40 °C (30-43 °C) below the ambient temperature to avoid wavelength drifts as well as to remove dark noise and readout noise. The following statement is added in the revised MS (page 4, lines 103-105): "Temperature of the used CCD-camera was controlled at 40 °C below ambient temperature to avoid emission spectrum drifts as well as to remove dark noise and readout noise."

14. Page 4, line 98, Was the spectral resolution measured? Where does it come from?

**Response** : The spectral resolution was determined by experimental measurement of emission line from a known light source, AS-363 Xenon lamp. The value of 0.59 nm was obtained for a width of 50 µm of the used CCD entrance slit.

15. Page 4, line 102, "semi-continuous", What is meant by semi continuous? This is rather unspecific.

**Response** : Here "semi-continuous" means that the instrument was developed to measure HONO during intensive field campaign or field deployment lasting from 1 to several months. Being based on an online chromatographic technique, it alternates a sequences of sampling periods and analytical period, but this has been described in detail in Afif et al, 2016 (cited) and is not the topic of the present paper. For clarity, we removed this term ("semi-continuous") from the revised manuscript.

16. Page 4, lines 114-115, "from integration of the peak and a calibration calculation." rather unspecific

**Response** : We modified this statement as follow in the revised manuscript (page 4, lines 121-124):

"The response obtained by integration of the chromatographic peak for the second stripping coil l is then subtracted from that of the first one to eliminate interferences. HONO concentrations are then calculated from this net signal using calibration factors determined through direct calibrations of the analytical system (HPLC-UV-Visible) performed using NaNO2 standard solutions."

17. Page 4, line 124, "effective", In a White cell this is not an "effective" but a "true" pathlength.

**Response** : Following reviewer suggestion, we deleted the term "effective" for White cell.

18. Page 4, line 126, "(see Fig. 1–insert)", The angle is not evident from Figure 1. Figure 1 needs to be improved in that regard.

**Response** : The figure 1 has been updated and an insert has been added to show a schematic view from the top showing the angle between the two in-situ spectrometric pathways.

19. Page 5, line 130, What are these detection limits based on? Absorption at specific individual wavelengths of the target species, or is it based on fits of bands in the recorded spectra?

**Response** : The detection limit is given as indicated value only. They are based experimentally on the fit and they take into account not only the specific absorption of the considered molecules but also the noise level on the condition of the experiments (quality of the alignment of the system, chosen integration

time …. as well as the presence of interfering species (e.g. water, $CO_2$ …. etc). Thanks to reviewer's remarks we have spotted that the detection limits was not related to the spectra window used in this study. We have hence modified the text accordingly.

20. Page 5, line 135, "catalyzed converter", The converter itself in not catalyzed. What are you trying to say?

**Response** : Agree with the reviewer, the statement is modified as: "via a heated molybdenum-converter,"

21. Page 5, line 148, What are the errors here?

**Response** : "The measurement error was 1% for RH and 0.1 °C for temperature at atmospheric pressure and room temperature." (page 5, lines 156-157) was added.

22. Page 5, lines 152-153, "it provides a 2 m long diameter", I am not getting this, I am afraid.

**Response** : the statement is revised as: "it provides a 2 m length as physical base length for both the FTIR's White cell and IBBCEAS' cavity"

23. Page 5, line 156, A schematic drawing of the position of the instruments would be helpful here.

**Response** : We updated figure 1 to give a better schematic drawing of the position of the instruments.

24. Page 6, lines 163-164, "When $NO_2$ concentration inside the chamber stabilized at 120±5 ppbv, water vapor produced in a small pressurize stainless steel vessel filled with ultrapure water (18.2 Mohm, ELGA Maxima).", this sentence does not make sense - check the English. "was filled" ???

**Response** : This sentence is revised (page 6, lines 172-174): "A pressurize stainless steel vessel filled with ultrapure water (18.2 Mohm, ELGA Maxima) was used to produce water vapor. When the $NO_2$ concentration inside the chamber was stabilized at 120±5 ppbv, $H_2O$ vapor was introduced into the simulation chamber."

25. Page 6, line 167, "a significant gas-phase HONO is systematically observed.", Improve the English

**Response** : "a significant gas-phase HONO is systematically observed." is changed to "a significant amount of gas-phase HONO is systematically produced for the present investigation work"

26. Page 6, line 168, "As described in the literature (Finlayson-Pitts et al., 2000; Lammel et al., 1995; Spataro et al., 2014).", This sentence is misplaced here it seems. Also improve the English.

**Response** : The statement is modified as: "As stated in the literatures (Finlayson-Pitts et al., 2000; Lammel et al., 1995; Spataro et al., 2014), HONO is generated through ….".

27. Page 6, lines 175-176, "was performed to allows the investigation of the sensitivity of the UVIBBCEAS data analysis to the interferences in the UV range.", English.

**Response** : "allows" was corrected to "allow": "was performed to allow the investigation on the sensitivity of the UV-IBBCEAS data analysis to the spectral interferences in the UV range".

28. Page 6, lines 180-181, "This process explains the peak shape formed of a straight injection step followed by an exponential decay of the various experiences.", This sentence is misplaced here it seems. Also improve the English.

**Response** : It's modified as: "This process explains the observed peak shape resulting from a straight injection step followed by an exponential decay during four-day experiments ($1^{st}$ to $4^{th}$ peaks in Fig.7 (a) and Fig. 8 (a))." And this sentence was moved to "Results and discussion" section on page 9, lines 273-275.

29. Page 6, line 188, certainly also older, more original IBBCEAS literature should be cited here.

**Response** : References "Gherman et al., 2008; Fuchs et al., 2010; Wu et al., 2012; Wu et al., 2014; Duan et al., 2018; Jordan et al., 2020" have been added into the revised manuscript on page 7, lines 198-199.

30. Page 7, line 190, The mirrors were not purged - why?

**Response** : Because our measurements were performed in a clean and particle-free chamber, the cavity mirror reflectivity $R$ ($\lambda$) was not significantly changed (within an uncertainty < 0.5%) which was checked before and after each experiment ($2^{nd}$ to $4^{th}$ peaks in Fig. 7(a) and Fig. 8(a)). If particles are present in the chamber, purging cavity mirror would be necessary to prevent contaminants on the mirrors.

31. Page 7, 197, "To do so", English

**Response** : "To do so" has changed to "In order for determination of $R$ ($\lambda$)"

32. Page 7, line 200, "the real cavity length $d$", Why would you retrieve the cavity length from a fit? That gives you one more fit parameter which might even correlate with some of the other "true" fit parameters. This is not meaningful in my opinion.

**Response** : "the real cavity length $d$" is not determined from a fit. This sentence is changed to "the mirror-to-mirror distance of the optical cavity $d$".

33. Page 7, line 211, "(number densities $n_{NO2}$, $n_{HONO}$, $n_{CH2O}$, a, b and c)", d is not mentioned here as fit parameter?

**Response** : "$d$" is the distance between two cavity mirrors (d=2.13 m) and is not a parameter to fit.

34. Page 7, line 212, What was the best fit range? 350-380 nm? Did you take the entire spectrum?

**Response** : We recorded the entire spectrum of 290-480 nm. But only the LED emission region of 351-378 nm were used for fit.

35. Page 7, lines 214-215, "during data recording with the software procedure which is comparable to the discussion on the IBBCEAS set-up evaluation", What are you trying to say here?

**Response** : The statement is modified as (Page 8, lines 230-232) : Acquisition time for each spectrum was 2 minutes, the statistical error of each individual spectrum is close to ~1%. This ~1% statistical error is as good as the results reported in the references for other IBBCEAS setups (Kleffmann et al., 2007; Fuchs et al., 2010; Varma et al. 2009; Gherman et al., 2008; Rodenas et al., 2013; Min et al., 2016).

36. Page 7, line 216, Varma et al. 2009 clearly missing here. Other original IBBCEAS literature and LOPAP literature in the context of chamber work is also not cited in this publication and should be included. e.g. Kleffmann and co-workers, Ruth and co-workers, Brown and co-workers ... Some related studies at CEAM (Valencia) and SAPHIR (Juelich) also seem to be missing.

**Response** : References including Kleffmann et al., 2007; Fuchs et al., 2010 (Brown and co-workers, SAPHIR, Juelich), Varma et al. 2009 (Ruth and co-workers); Gherman et al., 2008 (Ruth and co-workers) and Rodenas et al., 2013 (CEAM, Valencia) have been added at the right places in the updated manuscript.

37. Page 8, line 222, "351 to 378 nm", was that the fit range?

**Response** : Yes, spectral region of 351-378 nm was used to fit, when we calculated $1\sigma$ minimum detectable concentration (MDC) for HONO and $NO_2$, we used 362-372 nm residual data. But for $CH_2O$, the spectral data of 351-360 nm was used to estimated $1\sigma$ MDC. MDC (or DL) for $CH_2O$ should be 41 ppbv not 5 ppbv with 120 s. We have corrected this error. The corresponding text in section 2.3.1 page 8, lines 242-244 has been thus revised as follows:

"Based on the fit residual, the corresponding $1\sigma$ minimum detectable concentration (MDC) with mixing ratio for 120 s integration time are 112 pptv for $NO_2$, 56 pptv for HONO using 362-372 nm region data. MDC for $CH_2O$ with 120 s is 41 ppbv by using of 351-360 nm spectral data."

38. Page 8, line 225, The stated values are all mixing ratios by volume.

**Response** : We use the uniform way of "ppbv (part per billion by volume)" and "pptv (part per trillion by volume)" in the revised manuscript.

39. Page 8, line 229, "1 s per spectrum", The other way around: 1 spectrum per second

**Response** : correcting "1 s per spectrum" to "1 spectrum per second".

40. Page 8, line 233, Why is there no minimum observed in the Alan variance plot? Do the authors have an explanation for this? What make them select 120 s as integration time? Where is the "compromise"?

**Response**: In our investigated time range of 180 s, these linear Allan variance plot shows a White noise dominant regime for the developed IBBCEAS system. Using longer integration time (for example, 1000 s),

the minimum will be observed in the Alan variance plot. Here, as a compromise between detection limit (requiring long integration time) and measurement time response (requiring short measurement time), an integration time of 120 s was selected for use in the present work.

41. Page 8, line 235, What kind of apodization (if any) was used?

**Response** : The apodization function used is "Happ-Genzel". This has been added in the text

42. Page 9, line 255, "datum", ? data point?

**Response** : "datum" is changed to "spectrum"

43. Page 9, line 256, "response time", What is the "response time"? The integration time?

**Response**: "response time" corrected to " integration time "

44. Page 9, line 259, "committed", ?? what do you mean here. "executed"?

**Response**: "committed" changed to "performed".

45. Page 9, line 263, "positive", What is meant by a "positive interference". What is meant by "positive"?

**Response**: "positive" means $NO_2$ concentration measured by NOx analyzer is higher than the real $NO_2$ concentration, i.e. NOx analyzer overestimates $NO_2$ concentration.

46. Page 9, line 266, "After correction of the HONO contribution", How was this done. Was HONO measured at the same time? More detail is necessary here on the correction procedure. Is there an explanation as to why the times associated with the loss of $NO_2$ are time dependent for the NOx analyzer? What are the losses due to? Wall losses, or was the chamber continuously purged?

**Response**: The following sentences below are added in the revised MS to explain how this correction was done (page 10, lines 288-291): "The amount of the overestimated $NO_2$ concentration was attributed to HONO contribution that was simultaneously measured by NitroMAC. The real $NO_2$ concentration was then obtained by deduction of the HONO concentration measured by NitroMAC from the $NO_2$ concentration measured by NOx analyser."

The loss was mainly due to wall losses.

47. Page 9, lines 268-269, "Total intercomparison measurements", What is a "total intercomparison measurement". Total?

**Response**: "Total intercomparison measurements of $NO_2$ have been then compared" has changed to "Measurements of $NO_2$ have been then compared"

48. Page 9, lines 272 and 274, "98.65% and 88.47%", Should not be stated in %

**Response**: Using "0.987 and 0.885" to replace "98.65% and 88.47%" for correlation coefficient of $r^2$. All other similar data were also corrected to the decimal digits without %.

49. Page 9, line 273, ",", Full stop. Two sentences.

**Response**: Corrected

50. Page 10, lines 293-294, "this difference is close from the measurement errors.", close from? What are you trying to say here?

**Response**: "close from" was corrected to "close to".

51. Page 10, line 306, This is interesting and should be better quantified. The term "better agreement" is too vague here.

**Response**: The statement is modified as follows (page 11, lines 328-331) :

If the absorption cross section from another publication (Brust et al., 2000) was used to retrieve HONO concentration, all HONO concentrations in IBBCEAS will increase 23%, which equal to multiply a factor of 1.23 to the currently presented HONO concentrations in Fig. 8(a). In this case, good agreement (with a linear-fit slope approaching 1) is observed between the HONO concentrations measured by LED-IBBCEAS and NitroMAC, respectively.

52. Page 10, line 311, "<5%", The correlation is expressed through the slope and should be stated in that way.

**Response**: The related sentence was revised as follows (page 11, lines 332-335) :

The correlation and the regression analysis for the comparison between the FTIR and the IBBCEAS ($2^{rd}$-$4^{th}$ peaks) is given in Fig. 8(c), displaying a slope of 0.952 with a y-axis intercept of 0.250 ppbv and a $r^2$=0.89. HONO-concentration variation profile ($2^{rd}$-$4^{th}$ peaks in Fig. 8(a)) coincides well with each other between IBBCEAS and FTIR with a correlation slope close to 1. The discrepancy (<5%) is mainly due to the larger measurement uncertainty of HONO by FTIR.

53. Page 11, lines 327-328, "nm is not the highest for its sensitive measurement.", rephrase. State a value or make this more quantitative in another way.

**Response**: revised as (page 12, line 351) "the corresponding $CH_2O$ absorption cross section near 350 nm is not the maximal value in this region for its sensitive measurement".

54. Page 11, lines 331-332, "offers the ability of self-calibration", This needs re-phrasing. What is meant here by self-calibration.

**Response**: "offers the ability of self-calibration" is revised as (page 12, line 355-356) "offers the ability of self-calibration based on unique wavelength-dependent specific absorption intensity of the target molecules".

55. Page 11, line 333, An aspect not addressed in this paper, is the "interference" of aerosol on the retrieval. Fast changing aerosol concentrations (<2 min) are the biggest challenge for spectroscopic detection methods, especially in the near UV. For formaldehyde the influence of aerosol would strongly impinge on the quality of that data. This aspect should at least be mentioned when potential detection limits for other light sources are estimated. I understand that the influence of aerosol on the retrieval was not subject of this study, however, that is progress would have most scientific and/or technological merit.

**Response**: The following revised statement is provided to address aerosol interference issue (page 12, line 358-366) :

The present work in an atmospheric simulation chamber, with excellent measurements correlation on $NO_2$, HONO and $CH_2O$ between IBBCEAS and other well-established instruments, shows that the IBBCEAS technique offers the ability of self-calibration for simultaneously measuring concentrations of these three species with high precision without significant interference influence even if their absorption cross sections are overlapped. For its application to an uncontrolled environment, the interference resulting from the presence of aerosols, in particular, would degrade the performance of the IBBCEAS measurement which is an issue to be carefully addressed. Under harsh environmental conditions, additional approaches, such as purging high-reflectivity mirror, using particle filter to reduce aerosol absorption and scattering, could be associated to extend the IBBCEAS technique to field campaign (Wu et al., 2014; Duan et al., 2018; Jordan et al., 2020).

56. Page 11, line 335, A uniform way of addressing formaldehyde should be used throughout the manuscript: H2CO, CH2O, HCHO

**Response**: Corrected, we used the uniform way of "$CH_2O$" for formaldehyde.

57. Page 11, line 338, "intercomparison of all instruments were found to be in good agreement", "the intercomparison" itself is not in good agreement, but "the data" are in good agreement.

**Response**: This sentence was revised to (page 12, line 372-373) "The intercomparison of the measured data shows a good agreement on the temporal trends and variability in HONO, $NO_2$ and $CH_2O$."

58. Page 11, line 340, "systematic bias", Systematic errors were mentioned in correlation analysis.

**Response**: Agree with reviewer, "systematic bias" and the whole sentences (Exception of measurements near instrument detection limits, no evidence was found for any systematic bias in any of the instruments) was deleted.

59. Page 11, line 341, "positive", positive?

**Response**: "positive" means "overestimation" (Villena et al., 2012).

60. Page 11, line 342, English

**Response**: This sentence was revised to (page 13, line 375) "Due to positive interference, $NO_2$ concentration measured using NOx analyzer was corrected by deduction of HONO contribution"

61. Page 12, line 351, "measurements spatially relevant", "sounded volume.", what are you trying to say? probed?

**Response**: "measurements spatially relevant of sounded volume" has been modified to (page 13, line 384) "measurements that spatially depends on the probed volume".

62. Page 12, line 354, "without interference influence", This statement is much too strong. Of course there are issues surrounding IBBCEAS and aerosol were not even considered. 2 sentences. It also has .

**Response**: "without interference influence" was deleted ("high-precision" is enough). The statements have been made in 2 sentences.

63. Page 12, line 357, "self-calibration", ?? Not sure what is meant here.

**Response**: as stated in the MS (page 12 lines 355-356): the term "self-calibrations" is conventionally used in optical sensor that measures light absorption by the target molecule to infer the concentration using its specific absorption line intensity ("self-calibrations"), in comparison to the needs of complicated calibration process using "external" chemical solutions for wet chemistry-based analytical instrument.

64. Page 19, line 546, what is L2 and L3. Divergence of the LED. FTIR pathway not well shown. The photograph has no explanations in it. The mirrors are not labelled, the size is not mentioned ... The figure can be strongly improved.

**Response** : Figure 1 has been updated according to reviewers' suggestion, the following information has been added in the figure caption in the revised MS (page 21, lines 610-613): "Insert display a photograph of the set-up (left-top) and a schematic view from the top showing the angle between the two in-situ spectrometric pathways (left-bottom). L2 and L3 are BK7 achromatic focus lens. Cavity mirrors had 25 mm in diameter, 2 m radius of curvature and 6.35 mm thickness.".

65. Page 19, line 551, No labels in figure 2. What is what? Where is the spectrometer?

**Response**: New labels have been added into the new figure as below:

[Figure]

66. Page 21, line 554, Is the shape of the LED spectrum real, or is it affected by some optical filters?  Is the top of the LED spectrum really flat or is the spectrometer saturating? Were the zero air cross-sections measured or are they cited cross-sections from the literature? If the latter is the case the reference should be given?

**Response**: The shape of the LED emission is the real measurement result from a spectrometer, which is not affected by any optical filters and did not have any saturation of the spectrometer.

A reference (Miles et al., 2001) (Figure caption in figure 3) is provided into the revised manuscript for the used zero air cross-sections.

67. Page 21, line 557, The cross-section references should be cited here also.

**Response**: Reference of Miles et al. (2001) has been added in the revised manuscript (page 2, line 622).

68. Page 22, line 560, Too little information in the caption. What does "decomposed NO₂" and "decomposed HONO" mean? What measurements does this refer to?

**Response**: Decomposed $NO_2$ ($n_{NO_2} \cdot \sigma_{NO_2}(\lambda)$), HONO ($n_{HONO} \cdot \sigma_{HONO}(\lambda)$) and $CH_2O$ ($n_{CH_2O} \cdot \sigma_{CH_2O}(\lambda)$) spectra are provided to show their individual contribution to the experimentally measured unique spectrum (Kennedy et al., 2011), which can be obtained from the fit using Eq. 3:

$$\alpha(\lambda) = n_{NO_2} \cdot \sigma_{NO_2}(\lambda) + n_{HONO} \cdot \sigma_{HONO}(\lambda) + n_{CH_2O} \cdot \sigma_{CH_2O}(\lambda) + a\lambda^2 + b\lambda + c$$

The following statement has been added to the revised manuscript (page 8, lines 239-241):

"In order to well show individual absorption of each molecule, the decomposed spectra (Kennedy et al., 2011) associated with the corresponding fit for NO₂, HONO and CH₂O are shown in Fig. 4(a) (blue line), Fig. 4(b) (black line), and Fig. 4(b) (green line), respectively."

69. Page 23, figure 5, Is there an explanation why a minimum is not reached? Is the system so incredibly stable? This should be by volume, i.e. ppbv. Same on the other axis. axis labelling not uniform. ->

0.001. Typo in axis title! The green dashed line, being the chosen integration time, is not included here.

**Response**: "Typical Allan variance curves are plotted in Fig. 5, illustrating a highly desired white noise dominated system stability. As a compromise between detection limit (requiring long integration time) and measurement time response (requiring short measurement time), an integration time of 120 s was selected for use in the present work, which correspond to the measurement precision of 100 pptv for $NO_2$, 30 pptv for HONO and 40 ppbv for $CH_2O$."

The axis labelling and unit (ppbv) and typo on axis title were all corrected.

70. Page 24, figure 6, why are the "decays times" (wall loss rates) different between different injections of $NO_2$? what happened here?

**Response**: Because the introduced $H_2O$ vapor and $NO_2$ concentrations (Figure 6) are all different at each injection. Please check the $H_2O$ vapor concentration tracked by T&RH sensor in the following figure.

[Figure]

The following sentence is added in the revised manuscript (page 9, lines 271-272): "The peak concentrations of $H_2O$ vapor, measured by the T&RH sensor, were 1.85%, 1.50% and 1.65% for 2nd to 4th peaks, respectively."

71. Page 25, figure 7, 1). ppbv. 2). I would call this "Time" rather than "Date". 3). ppbv on all axes. 4). CL NOx analyzer not applied uniformly throughout the manuscript.

**Response**: All these points have been corrected.

72. Page 26, figure 8(c), I take it that the IBBCEAS error bars are not missing but are within the size of the symbol. Is the difference in error really that large at 10 s of ppbv of HONO? In relative terms this is possible. In absolute terms the FTIR was used to calibrate the mirror reflectivity. The absolute uncertainty should not be better than what the calibration delivered, which is limited by the FTIR.

**Response**: Agree with the reviewer: absolute $NO_2$ concentrations measured by FTIR were used to calibrate mirror reflectivity, the uncertainty in $NO_2$ concentration related to the FTIR measurements would be transferred to the concentration measurement of HONO, $NO_2$ and $CH_2O$ by IBBCEAS. We therefore improved the averaged times to about 100 scans (5 mins) to determine the mirror reflectivity so as to decrease the measurement uncertainty to about 7%.

73. All other issues referred to grammar, typo or English usage highlighted by blue symbols in the MS have been revised according to the reviewer's comments.

---

## Author Response (AR1)

**Dear Editors,**

Thanks very much for your time and efforts dedicated to review our submitted manuscript. We have diligently addressed all the concerns raised by three referees and we thank the reviewers for the pertinent remarks that allow us to improve the quality of our manuscript. Below we provide our detailed response to their comments one by one. We hope that the revised manuscript will satisfy the requirements of the journal AMT.

Kind regards,

Hongming Yi, Mathieu Cazaunau, Aline Gratien, Vincent Michoud, Edouard Pangui, Jean-Francois Doussin, and Weidong Chen

**Our responses to the reviewer's questions point by point are presented as below.**

**Reviewer 1**

RC1: 'Comment on amt-2021-19', Anonymous Referee #1, 22 Mar 2021

The article describes simultaneous measurements of NO2 and HONO generated in a simulation chamber experiments by several in-situ monitors with a focus on UV LED based incoherent broadband cavityenhanced absorption spectrometer (IBBCEAS) and its comparison against FTIR and chemiluminescent instruments. The experiments are well planned, and meticulous description of the instrumentation, measurements, intercomparisons and error analyses are provided. Well written, in a general sense, although small text corrections are sought. However, several aspects of the study need to be addressed before the paper can be accepted for publication. I suggest that the manuscript be majorly revised as per the comments listed below:

1. The diameter of the chamber is 2.13 m (page 3, line 88). Is it exactly the separation of cavity mirrors? Also, mirror radius of curvature is 2 m. Does radius less than mirror separation have any impact on cavity transmission or its stability?

**Response** : L=2.13 m is the exactly measured cavity mirror to mirror distance, and the mirror radius of curvature is R=2 m, which satisfy the condition of stable optical cavity configuration:  $0 \le (1-L/R)(1-L/R) \le 1$ . Therefore, the used optical cavity configuration did not have any impact on cavity transmission or its stability.

2. In section 2.3.1, it is mentioned that  $I_0$  is measured with nitrogen or dry (zero) air in the cavity. In the equation (1) for extinction coefficient, terms for Rayleigh scattering and Mie scattering are included. I think your target samples are trace quantities of NO2, HONO, H2O (vapor) and CH2O; all in gaseous state. When your  $I_0$  already has zero air how can there be a contribution from Rayleigh scattering of air and Mie scattering by particles just by introduction of calibration gases? If your  $I_0$  was measured in vacuum, then Rayleigh scattering by gases would be significant. I could not find any introduction of aerosol into your cavity either. Therefore, the equation (1) is misleading; also true for the equation (2) for mirror reflectivity retrieval where second term inside the parenthesis cannot be detected with dry air present in the  $I_0$  spectrum.

**Response** : we agree with the reviewer, we missed some extra description for Eq.1. So, we added the following sentences at lines 201-209 on page 7 in section 2.31:

"Here  $\alpha_{\text{Ray}}(\lambda)$ ,  $\alpha_{\text{Mie}}(\lambda)$  and  $\alpha_{\text{abs-particle}}(\lambda)$  are needed to consider for real atmospheric condition or openpath observation. In a particle-free environment,  $\alpha_{\text{Mie}}(\lambda)$  and  $\alpha_{\text{abs-particle}}(\lambda)$  can be neglected."

For Eq.2, we removed the term  $\alpha_{Ray-Zero air}$  in equation, and added the extra narratives in line 195 in section 2.31:

"Otherwise, low-concentration NO2 in air (<200 ppbv) was used for determination of mirror reflectivity  $R(\lambda)$ , the Rayleigh scattering coefficient by zero air  $\alpha_{\text{Ray-Zero air}}$  ( $\alpha_{\text{Ray}}(\lambda)$  in Eq.1) of ~10-8 cm-1 between 350 nm and 380 nm can be neglected:  $\alpha_{\text{Ray}}(\lambda)\approx 0$ , thus  $R(\lambda)$  can be determined by using a known-concentration NO2 sample as below:"

$$R(\lambda) = 1 - d\left(\alpha_{NO_2} \times \frac{I_{NO_2}(\lambda)}{I_{Zero air}(\lambda) - I_{NO_2}(\lambda)}\right)$$
(2)

3. In fact, if one looks at figure 3 it can be seen that the Rayleigh scattering cross section is 6 - 7 orders of magnitude smaller than the absorption cross sections of the calibration and target gases. Therefore, including zero air scattering cross section spectrum in the calculations makes no sense to me. I suggest to correct section 2.3.1 appropriately.

**Response** : In order to provide a generally usable expression, we keep the term of Rayleigh scattering cross section in figure 3 to indicate its contribution to the total absorption in Eq. 1. In the present work, it can be neglected as indicated by the reviewer. We removed  $\alpha_{Ray}(\lambda)$  in Eq. 2 in the revised version and added extra descriptions (see above).

4. On page 7, line 212, a singular value decomposition method seems to have been adopted for analyzing data. No elaboration or specifics about this method could be found in the article. It is suggested that the method is elaborated in detail or appropriate reference(s) be specified.

**Response** : Because this method has been widely used for concentration retrieval in IBBCEAS, two references about singular value decomposition method have been added on page 8, line 228:

Yi, H., Wu, T., Wang, G., Zhao, W., Fertein, E., Coeur, C., Gao, X., Zhang, W., and Chen, W.: Sensing atmospheric reactive species using light emitting diode by incoherent broadband cavity enhanced absorption spectroscopy, Opt. Express **24**, A781-A790, doi:10.1364/OE.24.00A781 (2016).

Varma, R. M., Venables, D. S., Ruth, A. A., Heitmann, U., Schlosser, E., and Dixneuf, S.: Long optical cavities for open-path monitoring of atmospheric trace gases and aerosol extinction," Appl. Opt. **48**, B159-B171, doi: 10.1364/AO.48.00B159 (2009).

5. How did you calculate the overall uncertainty from the individual uncertainties that you specified?

**Response** : the usual method to obtain the overall uncertainty  $u_o$  based on the individual uncertainties  $u_i$  is given as follows (Mathieu Rouaud, (2013), Probability, Statistics and Estimation: Propagation of Uncertainties in Experimental Measurement):

$$u_{o} = \sqrt{\sum_{i} u_{i}^{2}} \ (i=1,2,3,...).$$

6. Line 220 on page 8: It is mentioned that the statistical uncertainty from the fit was included in your error analysis. No mention was found on how this was calculated and what is the magnitude you obtained as uncertainty from the fit. How did you estimate this?

**Response** : The fit uncertainty was determined using statistic of  $1\sigma$  deviation of all fitted residuals.  $1\sigma$  deviation of the fitted residual reflected the LED intensity fluctuation and wavelength drift as well as fit error. Deduction of the uncertainty contribution from  $\Delta I/I$ , the statistically related uncertainty of all  $1\sigma$  deviations of the fitted residual is the uncertainty from fit. Based on the statistical uncertainty, this uncertainty is smaller than 0.5%.

The following statement was revised to address the uncertainty from the fit: "statistical uncertainty from the fit (<0.5%) (page 8, line 236)"

7. From figure 3 (b) it looks like HONO has two absorption peaks, the first one close to 355 is interfering with an absorption peak of  $CH_2O$ . In fact, after 357 nm, there is no influence (spectral interference) of  $CH_2O$  presence on NO2 and HONO. Interestingly, the LED has an order of magnitude less power below 357 nm. If it is just the NO2 and HONO measurements that the authors are interested in, use the spectral region from 360 nm would do the job. From figure 4 it is amply clear that in the 366 – 372 nm region, HONO and NO2 absorption can be clearly distinguished, then what is the need to extend it to 351 nm where light intensity itself goes negligible?

**Response** : We agree with the reviewer's pertinent analysis. 362-372 nm region is the best spectral range for high-sensitivity measurement of  $NO_2$  and HONO with stronger LED power and better mirror reflectivity under the present work condition that was not optimized for  $CH_2O$  measurement. The objective of this study was to investigate the possibility to make simultaneous measurements of these three species and their potential spectral interference (though the LED power was almost negligible), sensitive measurements of  $NO_2$  and HONO were performed in the 362-372 nm region.

8. While calculating the uncertainty, were the choice of spectral (sub) intervals used for analyzing different species considered? For example, like I said in the above comment, full spectral band or a sub interval of 360 - 378 can be used to analyze for NO2 and HONO. Would the results be the same? Could you get the same uncertainties for both spectral intervals? If different, then the error analysis must include the errors due to the choice of spectral interval. In your case, if CH2O is the target species, then analyzing it in the 351 - 360 nm band may be desirable than the full window (of course, with NO2 and HONO as co-analytes). If there is an effect due to spectral window choice, then optimization of the spectral interval of analysis for better results is desired.

**Response** : If the sub interval of 362 - 372 nm could be used to quantify NO2 and HONO, the detectable concentrations (MDC) would be improved to be 112 ppt and 56 pptv for NO2 and HONO, respectively. However, when using only the 351-360 nm spectral range to retrieve CH2O concentration, MDC for CH2O will increase from 14 ppbv (using 351-378 nm spectral) to 41 ppbv. Based on the fitted residual at the different wavelength ranges, the different 1 $\sigma$  residual values depending on wavelength range have been added into the Fig. 4(c) for MDC estimation, the corresponding text in section 2.3.1 page 8, lines 242-244 has been also revised. The corresponding statement on the MDC has been revised as follows:

"Based on the fitted residual, the corresponding  $1\sigma$  minimum detectable concentration (MDC) for 120 s integration time are 112 pptv for NO2 and 56 pptv for HONO using the 362-372 nm region data. MDC for CH2O with 120 s is 41 ppbv using the spectrum in the 351-360 nm region."

While the MDC is only 41 ppbv if the narrower spectral region of 351-360 nm was used to CH2O concentration retrieval due to weaker LED emission intensity in this range (blue curve in Fig. 2(a)).

9. In the section 2.3.2 on page 8, FTIR spectral analysis is described. Spectral fit in support of FTIR measurements, like corresponding to figures 3 and 4 for IBBCEAS instrument, needs to be shown.

**Response** : The FTIR data obtained here results from the analysis of 679 individual spectra. In order to minimize the effect on interfering species and following a similar approach to what is done in remote sensing, each of these spectra has been divided in small spectral windows in for which the fit was perform separately. We could show one or some of these windows for one of a few of these spectra, but this would not inform the reader about the robustness of the whole data analysis. Further, the HONO are sometime 20 times (or more) smaller than the water line. This would be a problem if the fit would involve wide absorption band – like it is the case in the UV domain – but here the spectra are highly structured which allow the fit procedure to have enough sensitivity to distinguish well between the various species. We have tried to produce the requested graph but, despite obvious mathematical sensitivity, the results are not spectacular enough to be evaluated with naked eyes.

Furthermore, the FTIR analytical procedure is not the central topic of the present paper as this technique is employed in simulation chamber since the mid-80's and routinely used in the LISA research group since 1997 (see, eg. Doussin et al, 1997). We believe that giving too many details about this part of the work would dilute the main message of the paper that is about the use of an in-situ CEAS system in a simulation chamber.

Some graphs have been produced (see below) but we don't think it is relevant to add them to the paper as we believe that it is more appropriate to keep the focus on the IBBCEAS development.

Figure 1 Spectral fit of the HONO bands in the 1200-1300 cm-1 region. The red \* in the first panel indicate a key feature of the HONO spectrum: the nu-3 band of the trans HONO-isomer at 1263 cm-1. Second and third panel is the Barney et al, 2001 spectrum scaled to 52 ppbv under our experimental condition and compared in the third panel to the experimental spectrum after water subtraction.

10. In this study authors used "synthetic" reference spectrum by Barney et al., 2001, for analyzing HONO from FTIR measurements. On page 10, line 303, the authors mention that the use of a different cross section for HONO introduced a larger error. Could you do an error analysis for FTIR measurements similar to that performed for IBBCEAS as well and add to the paper?

**Response** : What is meant here is that HONO is a complex molecule to handle in the lab because of heterogeneous formation as well as because of destruction processes. In consequence, the values of its absorption cross sections are still a matter for discussion as recalled by Gratien et al, 22009 (cited). Obviously, the use of imprecise cross sections leads to systematic errors. Our analysis of the state of the knowledge on this topic is that the work of Barney et al, 2001 is among the more reliable evaluation of the infrared cross section of nitrous acid.

In addition, the use of synthetic spectrum as the advantage - in comparison with the use of homemade reference spectrum – to avoid bringing more instrumental noise in the spectral fit which would increase the random errors.

Further, the risk of having traces of  $NO_2$  in an experimental reference spectrum is not negligible which can bring cross-estimation errors.

The FTIR in-situ use in chamber is quite well-known since a long time; It is known that we are working here at the limits of its capabilities and why it is interesting to develop new techniques such as IBBCEAS.

Having said this, again, we believe that the paper is more oriented toward the description of the IBB-CEAS system than the in situ FTIR and that a thorough error analysis for FTIR measurements is beyond the scope of this paper.

11. On page 9, line 274, comparison between IBBCEAS and FTIR measurements is specified. Although the discrepancy between the measurements is attributed to lower detection sensitivity of FTIR, the measurements of concentrations at well above detection limits show considerable deviations (Figure 7c). So, this is not the reason as mentioned in the paper. Either the FTIR is underestimating the NO2 concentrations due to some bias or the IBBCEAS is overestimating. Spectral fit from FTIR would help in this situation to assess the spectral validation of the measurements. I would tend to believe IBBCEAS measurements, more so because its calibrations and spectral validations are provided whereas less information regarding FTIR spectra and fit are provided in the paper. Error bars are given for IBBCEAS side, but no uncertainty analysis seems to be available for FTIR. This needs to be addressed better than attributing deviations to lower sensitivity of FTIR. Lines 339-340 in the section 4 (Conclusions) need to be corrected accordingly.

**Response** : Because of the presence of many strong water lines under the condition of the experiment in the 1500-1900 cm-1 region, the spectral region used to estimate NO2 concentration was 2830-2950 cm-1 which is far from exhibiting similar intense NO2 absorption. In consequence, the typical detection limit of 5 ppbv in this system were not relevant anymore (this has been corrected in the text). The standard

deviation due to the random errors is certainly rather in the range of 10 ppbv lead to and uncertainty of ca. 20 ppbv as can be confirmed from the plots of Fig. 7. When considering a random error of ca. 20 ppbv, the differences between IBBCEAS measurements and FTIR measurements become truly difficult to discuss.

This was not unexpected and this is why we relied on the chemiluminescence analyzer  $NO_2$  data (if corrected from HONO interferences) which fit well with IBBCEAS measurements.

12. This paper has both FTIR and IBBCEAS mounted on the wall of an atmospheric simulation chamber with two separate pieces (like a transmitter-receiver arrangement or the like) with atmosphere inbetween. This is done to directly use atmosphere as sampling volume and avoid sampling losses in extracting it into a containing volume. This is a step towards applying this technique into the real atmosphere. However, the real atmospheric conditions may have other interfering gases and suspended aerosol (absorbing, non-absorbing, SOA, etc.). Many atmospheric simulation chamber experiments assessing such measurement methodologies specify the instrument performances under varying atmospheric conditions. How would your measurement sensitivity be affected in the presence of aerosols in the chamber?

**Response** : Yes, you are right. In the case of aerosols present in the chamber, when the Mie scattering and absorption by particles,  $\alpha_{\text{Mie}}(\lambda)$  and  $\alpha_{\text{abs-particle}}(\lambda)$ , become similar to, or even higher than the mirror loss  $(1-R(\lambda)/d))$ , they are no longer negligible which might strongly shorten the wavelength-dependent effective cavity path length, and thus decrease the measurement sensitivity of the target species.

13. Finally, a few suggestions for text edits:

a. On page 2, line 43, "topics" to be replaced by "topic"

b. Line 52, "following" to be replaced by "followed"

c. On page 6, line 163. The sentence starting with "When NO2 concentration ...." is incomplete.

d. Line 168. The sentence starting with "As described ..." is incomplete.

**Response** : All these typo or grammar errors have been corrected. Thanks!

**Reviewer 2**

RC2: 'Comment on amt-2021-19', Anonymous Referee #2, 12 Apr 2021

The paper describes a IBBCEAS which, on the contrary to many existing set-ups, introduces the innovation of its in-situ installation, avoiding unwanted invasive use of pumps, etc. The system can measure HONO and, simultaneously, NO2 and CH2O. To evaluate its performance, an intercomparison against other instruments, NitroMAC, FTIR and NOx monitor is carried out. The paper is well written and results are well discussed. There is a detailed description of the instrumentation, procedures and error analysis. For these reasons I recommend its publication after considering the following aspects:

24: The title says HONO,  $NO_2$  and  $CH_2O$ , but the introduction mainly talks about HONO. HONO measurement is challenging, while the detection of  $NO_2$  and  $CH_2O$  is better stablished. Nevertheless, I would suggest to either include brief information on  $NO_2$  and  $CH_2O$  or explain that the main interest is

measuring HONO although  $NO_2$  and  $CH_2O$  absorb in the same region and are also tracked, being an advantage of the technique.

**Response** : We agree with the reviewer's opinion, the following sentences have been added in the introduction section (page 3, lines 69-72):

"Although the main interest for current work is to measure HONO, NO2 and CH2O are two other important atmospheric species (Washenfelder et al., 2016; Liu et al., 2020), these two molecules have strong absorption in the same region. Simultaneous measurements and quantification of HONO, NO2 and CH2O can be performed by the IBBCEAS techniques (Wu et al., 2014; Washenfelder et al., 2016; Duan et al., 2018; Jordan et al., 2020)."

80: This work introduces some changes in the set-up of the instrument, but it is based in previously developed IBBCEAS. Please, add some references.

**Response** : Four references related to previously reported IBBCEAS (Gherman et al., 2008; Fuchs et al., 2010; Wu et al., 2012; Wu et al., 2014; Duan et al., 2018; Jordan et al., 2020) have been added into section 2.3.1 of the revised manuscript (page 7, line 198-199):

There are only two references related to measurements of  $NO_2$  and HONO in simulation chamber (Gherman et al., 2008; Fuchs et al., 2010). Two recently published papers reporting on measurements of NO2 and HONO in ambient air have been added as references:

Duan, J., Qin, M., Ouyang, B., Fang, W., Li, X., Lu, K., Tang, K., Liang, S., Meng, F., Hu, Z., Xie, P., Liu, W., and Häsler, R.: Development of an incoherent broadband cavity-enhanced absorption spectrometer for in situ measurements of HONO and NO2, Atmos. Meas. Tech., 11, 4531–4543, doi:10.5194/amt-11-4531-2018, 2018.

Jordan, N. and Osthoff, H. D.: Quantification of nitrous acid (HONO) and nitrogen dioxide (NO2) in ambient air by broadband cavity-enhanced absorption spectroscopy (IBBCEAS) between 361 and 388 nm, Atmos. Meas. Tech., 13, 273–285, doi:10.5194/amt-13-273-2020, 2020.

225: Can you confirm that DL for  $CH_2O$  is 5 ppb? The emission of the LED below 356 nm is very low (Fig 3). The absorption for 143 ppb in Fig 4 doesn't seem to suggest that an absorption of 5 ppb will be detectable with such noise. DL has been calculated from 1- $\sigma$  in Fig 4 through the region 351-378 nm as it is the analysis region, but 1- $\sigma$  in the region where  $CH_2O$  absorbs is much higher, therefore, the real DL would be higher. That noise would also explain the noisy profile in Fig 9. Please, comment.

**Response** : Spectral region of 351-378 nm was used to fit, when we calculated  $1\sigma$  minimum detectable concentration (MDC) for HONO and NO2, we used 362-372 nm residual data. But for CH2O, the spectral data of 351-360 nm was used to estimated  $1\sigma$  MDC. MDC (or DL) for CH2O should be 41 ppbv not 5 ppbv with 120 s. We have corrected this error. The corresponding text in section 2.3.1 page 8, lines 242-244 has been thus revised as follows:

"Based on the fit residual, the corresponding  $1\sigma$  minimum detectable concentration (MDC) with mixing ratio for 120 s integration time are 112 pptv for NO2, 56 pptv for HONO using 362-372 nm region data. MDC for CH2O with 120 s is 41 ppbv by using of 351-360 nm spectral data."

580: There are -15 ppb of  $CH_2O$  in Fig. 9. It might be due to interference with HONO. On the one hand, in general, these unrealistic data can be withdrawn as they are below the DL. Indeed, those data seem to have been withdrawn from Fig. 9b since, looking at the 0 ppb of concentration for IBBCEAS, data for NITROMAC do not replicate the whole set of data in Fig. 9a, so they can be removed from Fig. 9a. On the other hand, they give information on how HONO is interfering, therefore, if the authors decide to include these data, some comment should be made in the text.

**Response** : Based on our updated analysis,  $1\sigma$  minimum detectable concentration (MDC) for CH2O is about 41 ppbv with 120 s, the data of about -15 ppbv of "CH2O" in Fig. 9 before the introduction of CH2O sample (without CH2O) are below the MDC, thus these unrealistic data before injection of CH2O have been withdrawn in the revised Figure 9a, as shown below. Because the measured CH2O concentration below 41 ppbv is not accurate. In the revised version, some unrealistic data have been withdrawn.

157: There were 4 experiments. At the beginning, the first experiment is described, and in line 174, it is said that there were 4 days of experiments. It can be mentioned that they were done under the same conditions as the first one.

**Response** : Yes, the 2th to 4th experiments were performed under the same experimental conditions as the first one. The procedures for 4 experiments were the same: firstly, the simulation chamber was pumped and evacuated to a pressure of ~ 1 mbar; secondly, the simulation chamber was filled with zero air to 1 atm (1000 mbar); and then, NO2 (<150 ppbv) sample was injected into the chamber for mirror reflectivity determination; finally, H2O vapor (<1.86%) were introduced into the chamber for HONO generation. During the whole experiment, IBBCEAS, NITROMAC, FTIR, NOx analyzer, temperature and relative humidity sensor (T&RH sensor), pressure sensor were running to record all related data for later analysis. The following description was added on page 6, lines 176-177 to describe the experiment procedure:

"There were 4 experiments during the whole measurement, the 2th to 4th experiments were performed under the same experimental conditions as the first one. The four experiments were followed by the same procedure."

172 and 198: Cavity mirror reflectivity is a key parameter in IBBCEAS for calculating the concentrations of the target molecules. Having a  $NO_2$  monitor, why did you use FTIR for its determination? The NOx

analyzer shouldn't have interferences during calibration as NO2 pure is introduced and there is no NOy (unless RH was not zero in the chamber). Is it related to accuracy? Please, add some comment.

**Response** : Because the RH in the chamber was not ideally zero, residual H2O vapor always existed inside the chamber, the estimated residual  $H_2O$  concentration is about 0.002% to 0.01% from T&RH sensor and FTIR. Once NO2 was introduced into the chamber, unknown-concentration NOy would be generated immediately. As discussed in the later section, positive interferences can't be avoided. So FTIR spectrometer was used to determine NO2 concentration for get more accurate mirror reflectivity in the present work.

240: Table 1 reflects the spectral regions corresponding to the IBI. Are these the analysis regions used for the analysis of each compound? If not exactly, please include this information in Table 1 or in the text.

**Response** : The table has been revised accordingly and the spectral windows used for the FTIR data analysis have been added.

255: Rephrase: "and 120 such acquisition data"... to "and 120 of such acquisition data" or "120 data acquired in this manner were"

**Response** : thanks reviewer for such helpful revision, correction has been done with "120 data acquired in this manner were".

276: detection limit of 10 ppbv at a sampling time of 1 min, compared .... (or similar, to distinguish from DL of 5 ppb at sampling time of 5 min in line 130).

**Response** : Agree with the reviewer, we corrected this typo error. The sentence is changed to "MDC of 10 ppbv at a sampling time of 5 min compared to that of 112 pptv in 2-min for IBBCEAS" (page 10, lines 301-302).

277: In Fig 7a, NO2 by FTIR is underestimated when there is CH2O. Was CH2O included as pure reference spectra in the analysis of NO2?

**Response** : First of all, we thank the reviewer for having spotted this. Indeed, because of the presence of many strong water lines under the condition of the experiment in the 1500-1900 cm-1 region, the spectral region used to estimate NO2 concentration was 2830-2950 cm-1. The choice of this region had two consequences:

- as the 2900 cm-1 region is far from exhibiting similar intense NO2 absorption, the random error from the NO2 quantitation was brought to ca. 20 ppbv.
- while CH2O was systematically included as pure reference spectra in the analysis leading to NO2, there seem to be an interference between the two species during the fit. We double check this